

# Implementation of the biogenic emission model MEGAN(v2.1) into the ECHAM6-HAMMOZ chemistry climate model. Basic results and sensitivity tests.

Alexandra-Jane Henrot [1,2], Tanja Stanelle [3], Sabine Schröder [1], Colombe Siegenthaler [4], Domenico Taraborrelli [1], and Martin G. Schultz [1]

[1]Forschungszentrum Jülich GmbH, IEK-8:Troposphere, Jülich, Germany.
[2]Unité de Modélisation du Climat et des Cycles Biogéochimiques, University of Liège, Liège, Belgium.
[3]ETH Zurich, Institute for Atmospheric and Climate Science, Zurich, Switzerland.
[4]ETH Zurich, Center for Climate System Modelling, Zurich, Switzerland.

*Correspondence to:* Alexandra-Jane Henrot (alexandra.henrot@ulg.ac.be)

**Abstract.** A biogenic emission scheme based on the Model of Emissions of Gases and Aerosols from Nature (MEGAN) version 2.1 (Guenther et al., 2012) has been integrated into the ECHAM6-HAMMOZ chemistry climate model in order to calculate the emissions from terrestrial vegetation of 32 compounds. The estimated annual global total for the simulation period (2000-2012) is 634 Tg C yr$^{-1}$. Isoprene is the main contributor to the average emission total accounting for 66 % (417 Tg C yr$^{-1}$), followed by several monoterpenes (12 %), methanol (7 %), acetone (3.6 %) and ethene (3.6 %). Regionally, most of the high annual emissions are found to be associated to tropical regions and tropical vegetation types.

In order to evaluate the implementation of the biogenic model in ECHAM-HAMMOZ, global and regional BVOC emissions of the reference simulation were compared to previous published experiment results with the MEGAN model. Several sensitivity simulations were performed to study the impact of different model input and parameters related to the vegetation cover and the ECHAM6 climate. BVOC emissions obtained with the biogenic model are within the range of previous published estimates. The large range of emission estimates can be attributed to the use of different input data and empirical coefficients within different setups of the MEGAN model. The biogenic model shows a high sensitivity to the changes in plant functional type (PFT) distributions and associated emission factors for most of the compounds. The global emission impact for isoprene is about -9 %, but reaches +75 % for $\alpha$-pinene when switching to PFT-dependent emission factor distributions. Isoprene emissions show the highest sensitivity to soil moisture impact, with a global decrease of 12.5 % when the soil moisture activity factor is included in the model parameterization. Nudging ECHAM6 climate towards ERA-Interim reanalysis has impact on the biogenic emissions, slightly lowering the global total emissions and their interannual variability.

## 1 Introduction

The majority of volatile organic compounds emitted from the terrestrial biosphere (BVOCs), including hydrocarbons (isoprene, monoterpenes, and sesquiterpenes) as well as oxygenated organic compounds, are highly reactive and have been shown to affect both gas phase and heterogeneous atmospheric chemistry at local and global scales (Ashworth et al., 2013). Photo-oxidation



of BVOCs notably, in the presence of nitrogen oxides (NOx), contributes to the formation of carbon monoxide (CO), hydroxyl radical (OH) and tropospheric ozone (Pfister et al., 2008; Granier et al., 2000), thus influencing the oxidative capacity of the atmosphere (Atkinson and Arey, 2003; Pacifico et al., 2009; Taraborrelli et al., 2012), and leads to secondary organic aerosol (SOA) particle formation (Hallquist et al., 2009; van Donkelaar et al., 2007). Through their effects on atmospheric chemistry, aerosol concentrations, and the global carbon cycle, BVOC emissions as well influence global climate (Constable et al., 1999; Collins et al., 2002). Terrestrial vegetation is thought to account for around 90 % of the total of non-methane VOCs emitted into the atmosphere each year (Guenther et al., 1995). Isoprene is quantitatively the most important of the BVOCs, with an estimated global annual emission of about 400-600 Tg of carbon (Arneth et al., 2011; Guenther et al., 2012).

BVOCs are therefore a crucial component of the Earth system that has to be considered in global and regional chemical transport models. Quantitative estimates of their emissions into the atmosphere are needed for numerical assessments of their impacts on past, present and future air quality and climate (Sindelarova et al., 2014). BVOC emission strengths vary by plant species and depend on biological parameters (e. g. water stress) (Pfister et al., 2008; Pegoraro et al., 2004), physical conditions (e. g. temperature, radiation) (Guenther et al., 1995; Li and Sharkey, 2013), and chemical variables (e. g. tropospheric ozone, carbon dioxide) (Velikova et al., 2005; Rosenstiel et al., 2003). Several models have been developed for estimation of BVOC emissions from vegetation (Pierce and Waldruff, 1991; Guenther et al., 1995; Niinemets et al., 1999; Martin et al., 2000; Arneth et al., 2007).

The biogenic emission model used in the present study is based on the MEGAN model (Guenther et al., 1995, 2006, 2012). The current version of MEGAN, MEGANv2.1 (Guenther et al., 2012), simulates the fluxes of 20 classes of BVOCs which are then decomposed into 150 individual species, including isoprene, monoterpenes, sesquiterpenes, and other oxygenated volatile organic compounds. During the past ten years, MEGAN has been widely used within the scientific community for the estimation of BVOC emissions as an offline model (Guenther et al., 2006, 2012; Müller et al., 2008; Sindelarova et al., 2014; Messina et al., 2015) and has been incorporated into various earth system and chemistry transport models (Guenther et al., 2006, 2012; Heald et al., 2008; Pfister et al., 2008; Stavrakou et al., 2009; Emmons et al., 2010; Tilmes et al., 2015; Messina et al., 2015)(Guenther et al., 2006; 2012; Heald et al., 2008; Pfister et al. 2008; Stravakou et al., 2009; Emmons et al. 2010, Tilmes et al., 2015; Messina et al., 2015).

In this study, we have implemented the MEGANv2.1 model into the ECHAM6-HAMMOZ chemistry climate model. The aim of the present study is (i) to present the updated version of the biogenic emission module implemented into ECHAM6-HAMMOZ, (ii) to evaluate present-day global and regional scale emissions for a series of 32 compounds, (iii) to compare MEGAN-ECHAM-HAMMOZ basic results to previous offline and online experiment results with the MEGAN model and (iv) to test the sensitivity of BVOC emissions to climate and vegetation dependent model parameters.





## 2   Model description

### 2.1   Atmospheric model ECHAM-HAMMOZ

ECHAM-HAMMOZ is a comprehensive chemistry climate model that describes aerosol and gas-phase chemical processes in the troposphere and stratosphere including their coupling via heterogeneous reactions, and oxidation of aerosol precursors.
ECHAM-HAMMOZ is developed by a consortium composed of ETH Zurich (Switzerland), Max Planck Institute for Meteorology (Germany), Forschungszentrum Jülich (Germany), University of Oxford (UK), Institut für Troposphärenforschung (Germany) and the Finnish Meteorological Institute (Finland). We used here the most recent version ECHAM6.3.00-HAM2.3-MOZ1.0(rc2). The model is based on the ECHAM6 atmospheric general circulation model (Roeckner et al., 2003; Stevens et al., 2013), including the phenology model JSBACH (Raddatz et al., 2007; Brovkin et al., 2009), the HAM aerosol model (Stier et al., 2005; Zhang et al., 2012), and a chemistry module derived from the MOZART chemistry transport model (Emmons et al., 2010; Lamarque et al., 2012). For the simulations of this study, ECHAM6-HAMMOZ is employed in its stand-alone atmospheric GCM mode, i.e. without interactive aerosol and chemistry. Thus, we basically run the ECHAM6 GCM with the MEGAN emissions module as the sole chemistry component. ECHAM6 dynamics (vorticity and divergence of the wind field, temperature and surface pressure) are calculated in spectral space with triangular truncation at term 63 (T63), while physics are calculated on a 1.8 x 1.8 Gaussian grid (Roeckner et al., 2003). The simulations use 47 vertical levels, from the surface to 0.01 hPa, and a time step of 7.5 minutes. Sea-surface temperatures and sea-ice coverage are prescribed for each year of simulation, following the Coupled Model Intercomparison Project Phase 5 (CMIP5) AMIP-simulation protocol (Giorgetta et al., 2012). The gas climatologies of $CO_2$, $CH_4$, $N_2O$, and chloroflourocarbons (CFCs) are specified by a single value meant to be representative for the tropospheric concentration of each year of simulation. For several sensitivity simulations, ECHAM6 is run in nudged mode, constraining large-scale meteorology by the ECMWF ERA-Interim meteorological fields. The nudging tendency is applied after model dynamics, in spectral space. The nudging time scales are 6h for vorticity, 48h for divergence, 24h for temperature, and 24h for surface pressure (Lohmann and Hoose, 2009; Zhang et al., 2014).

### 2.2   Land surface model JSBACH

JSBACH is a state-of-the art Earth System Model land surface scheme that simulates fluxes of energy, water, momentum, and $CO_2$ between land and atmosphere, including interactive and dynamic vegetation (Raddatz et al., 2007; Brovkin et al., 2009). The modeling concept of JSBACH is based on a tiled (fractional) structure of the land surface. Each land grid cell is divided into tiles covered with a variable number of Plant Functional Types (PFTs), bare surface, and tiles with land cover excluded from natural vegetation dynamics (Reick et al., 2013). The soil hydrology and temperatures are modeled by a five-layer scheme (Hagemann and Stacke, 2015). The dynamic vegetation component, simulating natural changes in biogeography on the basis of competition between PFTs (Reick et al., 2013), has not been activated in this study. The spatial distribution of the PFTs is prescribed on the basis of global potential land cover maps (Pongratz et al., 2008). For the present study, we used JSBACH with 11 PFTs, as fixed for the CMIP5 simulation protocol (Reick et al., 2013; Brovkin et al., 2013).





## 2.3 Biogenic emission module MEGAN

Emissions of biogenic compounds from terrestrial vegetation are estimated using the Model of Emissions of Gases and Aerosols from Nature MEGAN (Guenther et al., 1995, 2006, 2012). The current version of the model, MEGANv2.1 (Guenther et al., 2012), calculates the net primary emission of 20 compound classes, which are then decomposed into 150 individual species

such as isoprene, monoterpenes, sesquiterpenes, carbon monoxide, alkanes, alkenes, aldehydes, acids, ketones and other oxygenated VOCs. The net emission rate (in units of $\mu$g compound grid cell$^{-1}$ h$^{-1}$) of each compound into the above-canopy atmosphere from a model grid cell is calculated according to:

$$Emission(i) = EF(i) \cdot \gamma \cdot S \qquad (1)$$

where $EF(i)$ ($\mu$g m$^{-2}$ h$^{-1}$) is the emission potential (also named emission factor) of a compound $i$ into the canopy at standard

conditions of light and temperature ( i. e. photosynthetic photon flux density of 1000 $\mu$mol m$^{-2}$ s$^{-1}$ and leaf temperature of 30°C), $\gamma$ is the dimensionless emission activity factor that accounts for emission response to meteorological and phenological conditions, and $S$ (m$^2$) is the grid cell area.

The biogenic emission module implemented into ECHAM6-HAMMOZ is adapted from MEGANv2.1. It includes 32 compounds grouped into 17 classes (see Table 1).

### 2.3.1 Emission activity factor $\gamma$

The emission activity factor $\gamma$ for each compound is calculated following the MEGANv2.1 parameterization (Guenther et al., 2012):

$$\gamma = \gamma_{CE} \cdot \gamma_A \cdot \gamma_{SM} \cdot \gamma_{CO2}. \qquad (2)$$

$\gamma_{CE}$ accounts for variations associated with Leaf Area Index (LAI) (m$^2$ m$^{-2}$), Photosynthetic Photon Flux Density (PPFD)

($\mu$mol of photons in 400-700 nm range m$^{-2}$ s$^{-1}$) and temperature (K):

$$\gamma_{CE} = \gamma_{LAI} \cdot ((1 - LDF) \cdot \gamma_{TLI} + LDF \cdot \gamma_P \cdot \gamma_{TLD}. \qquad (3)$$

The activity factors for LAI ($\gamma_{LAI}$), light ($\gamma_P$) and temperature ($\gamma_{TLI}$ and $\gamma_{TLD}$) are calculated using the Parameterised Canopy Environment Emission Activity (PCEEA) approach of the MEGAN model (Guenther et al., 2006). We refer the reader to the description of Guenther et al. (2006) for the details of computation. The activity factor for temperature is divided

into the light dependent ($\gamma_{TLD}$) and light independent ($\gamma_{TLI}$) factor using the light dependence fraction $LDF$ specific for each compound (Guenther et al., 2012). The light dependent factor $\gamma_{TLD}$ is calculated following the isoprene-response to temperature described by Guenther et al. (2006). The light independent factor $\gamma_{TLI}$ follows the monoterpene exponential temperature response described by Guenther et al. (1993). Detailed formula and parameters per compound classe are given in Supplementary Material (Sect. S1 and S2).

The activity factors are calculated on the basis of the leaf area index, the lowest atmospheric model level temperature and surface photosynthetically active radiation at each time step in the ECHAM6-HAMMOZ model, as well as the average





temperature and radiation conditions over the last 24 hours. The leaf area index is calculated at each model time step in JSBACH taking into account a full plant phenology scheme, and is averaged over the vegetated part of the grid cell to be used in the biogenic emission module.

The $\gamma_A$ factor represents the leaf age emission activity factor. Its calculation is based on a decomposition of the canopy
into fractions of new, growing, mature and old foliage derived from the current and previous month LAI, following the parameterization described by Guenther et al. (2006, 2012). The $\gamma_{SM}$ and $\gamma_{CO2}$ factors account for the dependence of isoprene emission on respectively the soil moisture and the atmospheric concentration of $CO_2$ as described by Guenther et al. (2012). For compounds other than isoprene, $\gamma_{SM}$ and $\gamma_{CO2}$ equal 1. The soil water activity factor is evaluated using the relative soil water amount calculated with the soil water model included in ECHAM6 (Hagemann and Stacke, 2015). The atmospheric
$CO_2$ concentration is prescribed annually, using a global value from the Representative Concentration Pathway scenario RCP 4.5 (stabilization scenario where total radiative forcing is stabilized before 2100 (Thomson et al., 2011). By default, $\gamma_{SM}$ and $\gamma_{CO2}$ are not activated, and set equal to 1 for isoprene.

### 2.3.2 Emission factor $EF$

Emission factor $EF$ for each compound can be specified from global gridded potential emission maps based on species compo-
15 sition and species-specific emission factors compiled from detailed land cover and plant species distributions (Guenther et al., 2012). Emission factors from global maps are available from the original MEGANv2.1 code for 10 predominant compounds, i. e. isoprene, $\alpha$-pinene, $\beta$-pinene, 3-carene, limonene, myrcene, t-$\beta$-ocimene, sabinene, 232-MBO and nitric oxide. Another option to obtain the emission factor for each compound is to use plant functional types (PFT) distributions and PFT specific emission potentials (Guenther et al., 2012). The emission factor of a grid cell for each compound is calculated as:

$$EF(i) = \sum \epsilon(i,j) \cdot PFT_j \qquad (4)$$

where $\epsilon(i,j)$ is the emission factor of compound $i$ at standard conditions of light and temperature for the plant functional type $j$ (constant in time and space), and $PFT_j$ is the fraction of the grid cell covered by the PFT $j$. MEGANv2.1 includes a 15-PFT distribution derived from the PFT scheme of the Community Land Model version 4 CLM4 (Lawrence et al., 2011), related to the year 2000 and based on Moderate Resolution Imaging Spectroradiometer (MODIS) land surface datasets (Lawrence and
25 Chase, 2007) and a crop dataset (Ramankutty et al., 2008). Specific emission factors for each compound attributed to the 15 PFTs can be found in Sect. S3 of Supplementary Material. By default, the biogenic emission module runs with the emission factors from global maps for the 10 compounds listed above and calculates the emission factors from the PFT specific values and fractions for the other modelled compounds. Maps of emission factors as well as PFT fractions have been interpolated from their original resolution (0.5 x 0.5) in the current resolution of the ECHAM-HAMMOZ model (T63).
We have introduced in the biogenic emission module the possibility to replace the MEGAN2.1-CLM4 PFT distribution by the PFT distribution of the JSBACH model. In order to have a sufficiently detailed PFT classification to take into account the variability in PFT specific emission factors the JSBACH 11 PFT-classification is extended to 14 PFTs to be as close as possible to the MEGAN2.1-CLM4 15-PFT classification. The correspondence between both PFT classification is given in Table 2.



The extra-tropical tree types of the JSBACH 11 PFT-classification (i.e. extra-tropical evergreen (type 3) and extra-tropical deciduous (type 4)), as well as the C3 grass type are subdivided in order to make a distinction between broadleaf and needleleaf trees and temperate and boreal types. Fractions of broadleaf and needleleaf trees are derived from natural vegetation distributions (Pongratz et al., 2008, 2009), reconstructed from the global potential vegetation maps of Ramankutty and Foley (1999).

The separation between boreal and temperate trees, as well as cold and cool C3 grasses is based on bioclimatic limits taken from Sitch et al. (2003) for the trees, and from Oleson et al. (2010) (based on Levis et al. (2004)) for C3 grasses. Tree types are classified as boreal if the temperature of the coldest month (calculated from a 20-year run (1990-2009) with the ECHAM6 model) is below -2°C. C3 grasses are classified as cold or arctic (using the denomination of the MEGAN2.1-CLM4 PFT classification) if the temperature of the coldest month decrease below -17°C.

Tropical tree types (PFTs 1 and 2 in JSBACH) are not subdivided as their correspondence to MEGAN2.1-CLM4 tropical PFTs is straightforward. JSBACH shrub PFTs represent deciduous shrubs in warm/temperate and cold/boreal environments, and directly correspond to the shrub types of the MEGAN2.1-CLM4 classification. No evergreen shrub type is considered in JSBACH. In consequence, the broadleaf evergreen temperate shrub type (PFT 9) of the MEGAN2.1-CLM4 classification has no correspondance in the extended JSBACH classification. Finally, as MEGAN2.1-CLM4 PFT classification does not
distinguish grasses and pastures, these two types are clustered together in the extended JSBACH classification.

The PFT correspondance applied here enables the use of MEGAN2.1 PFT specific emission factors, calculated from global averages of species specific emission factors from more than 2000 different ecoregions (Guenther et al., 2012). The extended JSBACH-PFT classification is only used here in the biogenic emission module for the calculation of emission factors, and does not impact the calculation of the PFT fractions or any other calculations in the JSBACH model.

## 3    Simulation design

In order to evaluate the implementation of MEGAN in ECHAM-HAMMOZ and its sensitivity to different model settings, we have performed several simulations at the global scale at a T63 spatial resolution (listed in Table 3). All simulations have been run for 12 years for the period 2000-2012, starting from a one-year spin-up with ECHAM-HAMMOZ. For the reference simulation (CTRL), the biogenic emission module is run with the emission factors from global maps for 10 compounds (isoprene,
$\alpha$-pinene, $\beta$-pinene, 3-carene, limonene, myrcene, t-$\beta$-ocimene, sabinene, 232-MBO and nitric oxyde) and with MEGAN2.1-CLM4 PFT-specific emission factors for the rest of the compounds. In the first sensitivity simulations, we evaluate the impact of PFT distributions on the modeled emissions, by using the PFT-specific emission factors calculated from the PFT fractions taken from the MEGAN2.1-CLM4 (experiment PFT-CLM4) and the extended-JSBACH PFT (experiment PFT-JSBACH) distributions, respectively. In experiment TEST-SM, the impact of soil moisture on isoprene emissions is analysed, by activating
the soil moisture activity factor. The effect of a constrained meteorology on the emissions simulated with the biogenic module are evaluated in the last experiments TEST-NUDG and TEST-NUDG+SM.



## 4 Biogenic emission model results

### 4.1 Reference simulation

The biogenic module monthly outputs have been averaged over the 2000-2012 period to calculate the global annual emission totals of the 32 compounds for the reference simulation (see Table 4). The global annual emission total for the averaged period

reaches $634.1\pm12.5$ Tg C yr$^{-1}$. Isoprene is the most emitted compound with a global annual emission of $417\pm10.2$ Tg C yr$^{-1}$ (473 Tg (species) yr$^{-1}$), accounting for 65.8 % of the total BVOC emissions. Monoterpenes global annual emission (considering only the sum of $\alpha$-pinene, $\beta$-pinene, limonene, sabinene, myrcene, 3-carene and t-$\beta$-ocimene emissions) is $78.3\pm1.2$ Tg C yr$^{-1}$ (88.8 Tg (species) yr$^{-1}$) and contributes to 12.3 % of the total BVOC emission. Among the monoterpenes, $\alpha$-pinene is the most abundant compound, followed by $\beta$-pinene and t-$\beta$-ocimene. Following the monoterpenes, methanol, acetone and

ethene are the most emitted compounds with respectively $43.9\pm0.5$, $22.9\pm0.3$ and $22.9\pm0.4$ Tg C yr$^{-1}$, contributing to 6.9, 3.6 and 3.6 % of the total emission. The carbon monoxide global annual emission is $41.1\pm0.6$ Tg C yr$^{-1}$.

Annual emissions of the five most emitted compounds and $\beta$-caryophyllene for seven regions of the world are presented in Fig. 1. Regions have been defined following the protocol of the GlobEmission project (http://www.globemission.eu), except that African regions have been grouped into one. All the compounds are mainly emitted in south and tropical regions with a

15 dominance of South America, followed by Africa, South Asia and Australia. In the Northern Hemisphere, emissions mainly come from North America, followed by Russia and Europe.

Global, Northern and Southern Hemisphere monthly mean emissions for the six compounds averaged over the reference simulation period are presented in Fig. 2. The maximum global emission occurs in July for most of the compounds and emissions are generally higher during the Northern Hemisphere summer. Regions of the Northern Hemisphere are minor

contributors to the global annual total emissions, but seasonal emissions fluctuations in the Northern Hemisphere mainly impact the global seasonal emission profile. Isoprene shows a more constant seasonal cycle, due to the fact that it is mainly emitted by tropical regions with opposite seasonal cycles, thus compensating in the global annual mean.

Spatial distributions of the main compound emissions for the Northern Hemisphere summer (June-July-August emission mean) and winter (December-January-February emission mean) are shown in Fig. 3 and Fig. 4. As observed in the global

regional means (Fig. 1), emissions are generally higher in tropical regions, especially South America, Africa and Indonesia and are lower at middle and high latitudes. Tropical regions are high emitters due to the year-long warm temperatures and high incoming radiation, together with high biomass density and emission factors associated to tropical vegetation types. In the temperate and high latitude zones dominated by deciduous and coniferous forests, emissions vary over the seasons by several orders of magnitude due to seasonal fluctuations in temperature and solar radiation. The summer-winter contrast in the

Northern Hemisphere emissions is particularly marked for monoterpenes and methanol, for which temperate and coniferous vegetation types have high emission factors.



## 4.2 Comparison to previous emission totals calculated with the MEGAN model

In order to evaluate the basic results of the biogenic emission module in ECHAM-HAMMOZ, we compare the global and regional total emissions of the reference simulation with emission totals obtained from previous published experiments with the MEGAN model (listed in Table 5). The annual global isoprene emission in our reference simulation of 417 Tg C year$^{-1}$

is within the range of previous reported values calculated with different versions of the MEGAN models, varying between 361 and 601 Tg C year$^{-1}$. It is slightly less than previous estimates with different configurations of MEGANv2.1. (Messina et al., 2015; Sindelarova et al., 2014; Guenther et al., 2012). The annual global emission of monoterpenes is in general lower than reported estimates with MEGANv2.1, but closer to previous emission totals with MEGANv2 (O'Donnell et al., 2011; Emmons et al., 2010). It should be noticed that differences in monoterpenes global emissions may be due to the consideration

of different species in the global totals. Here, we simulate the emissions of seven monoterpenes, i.e. $\alpha$-pinene, $\beta$-pinene, limonene, sabinene, myrcene, 3-carene and t-$\beta$-ocimene emissions. Methanol and acetone emissions, as well as the rest of the compound emission totals presented in Table 5, are in the middle of the ranges of previous emission estimates. Ethene total emission is similar to the total reported in Guenther et al. (2012), but $\beta$-caryophyllene total emission is lower. Similarly to Sindelarova et al. (2014) we estimate a global biogenic emission of 1.4 Tg C year$^{-1}$ of toluene, being 18 % of the total

estimated anthropogenic emissions 7.6 Tg C year$^{-1}$ from the EDGAR database (Crippa et al., 2016). (Misztal et al., 2015) estimate a maximum biogenic source of 5.5 Tg C year$^{-1}$. Such a biogenic source would reduce the underestimate of toluene by atmospheric models (Cabrera-Perez et al., 2016).

     Many factors can explain the discrepancies between the reference simulation results of the present study and previous reported global estimates with the MEGAN model. First, each study uses its own set of input data (temperature, radiation, LAI,

land cover, etc), derived from atmospheric model outputs or reanalysis data. Time periods of the simulations also cover different years, from one single-year (Millet et al., 2010; Guenther et al., 2012) to several decades (Arneth et al., 2011; Sindelarova et al., 2014). As already pointed out by Arneth et al. (2011), and confirmed by the estimations of Messina et al. (2015); Sindelarova et al. (2014), different meteorological forcings can lead to significant differences in emission estimates, e. g. more than 10 % difference in isoprene total emission using MEGANv2 (Guenther et al., 2006; Arneth et al., 2011). Furthermore, spatial

resolutions of models and meteorological inputs vary over a large range, e. g. from a coarse resolution of 2.8° x 2.8° (Emmons et al., 2010) to a finer resolution of 0.5° x 0.5° (Guenther et al., 2012). The resolution effect can influence the total global emissions by a few percent, but can be more important if associated with coarse resolution land cover data (Pugh et al., 2013).

     BVOC emissions are also very sensitive to changes in land cover and LAI inputs. Global isoprene emission sensitivity to differences in LAI varies from less than 10 % in MEGANv2.1 (Sindelarova et al., 2014; Messina et al., 2015) to about 30 % in

MEGANv2 (Pfister et al., 2008; Guenther et al., 2012), depending on the LAI dataset considered. Messina et al. (2016) report a much lower sensitivity of other BVOC global emissions to LAI in MEGANv2.1 due to the specific light independent emission parameterization of the MEGAN model, but a significant response of the seasonal cycle of emissions to LAI changes. Changing the land cover or PFT distributions and emission factors results in a wide range of BVOC estimates. Pfister et al. (2008) find a difference of 24 % in isoprene global emission by changing PFT distributions in MEGANv2. Sindelarova et al. (2014) and



Messina et al. (2015) investigated the sensitivity of isoprene emission in MEGANv2.1 to different values of emission factors. The sensitivity of BVOC emissions to various PFT distributions and emission factors is addressed in the next section.

The use of different versions of the MEGAN model in the listed studies, with different parameterizations and empirical coefficients, can also result in differences in the global emission totals (Arneth et al., 2011). The introduction of light-dependent factors for other compounds than isoprene in MEGANv2.1 has notably a significant effect on global emissions (Messina et al., 2015). The activation of the $CO_2$ and soil moisture activity factors in MEGANv2.1 also impact isoprene global emissions. Taking into account the soil moisture effect decreases the global annual isoprene emission, by 7 % (Guenther et al., 2012) to 50 % (Sindelarova et al., 2014) depending on the soil moisture database. Similarly to Sindelarova et al. (2014), we have not activated the soil moisture factor for the reference simulation. It's impact on isoprene emission is described in the next section. When activated, the $CO_2$ activity factor leads to a slight increase of global isoprene emission by about 3 % for present-day emissions (Heald et al., 2009; Sindelarova et al., 2014). The $CO_2$ activity factor is normalized to 1 for an ambient $CO_2$ concentration of 400 ppmv and decreases non-linearly if $CO_2$ concentration increases (Heald et al., 2009). Thus, it has only a slight impact on present-day isoprene emission, but it has to be taken into account for future simulations with $CO_2$ concentration exceeding 400 ppmv. The $CO_2$ activity factor has not been activated in the reference simulation. It's activation increases the global isoprene emission by 2.5 %.

At the regional scale, the annual totals (Fig. 1) are very similar to the totals reported by Sindelarova et al. (2014) for isoprene and Messina et al. (2015) for isoprene and other compounds. Highest emissions occur in South America, Africa, South Asia, Australia and North America. The similarity in the spatial distribution of emissions results from the use of the same emission factor distributions (obtained from global emission factor maps and MEGAN-CLM4 PFT fractions) in the three studies. This result confirms that in the MEGAN model the spatial emission distribution mainly depends on the emission factor and PFT distributions, since the meteorological drivers are different between these studies.

Seasonal variations of the emissions of isoprene, monoterpenes and other compounds are also in line with the results of Messina et al. (2015), despite the use of different LAI distributions. Both results agree on a net maximum of emission in July for all the selected compounds, except isoprene which has a more constant seasonal emission profile with higher emissions in March and July and lower emissions in June. Thus, the use of the LAI distribution calculated with the JSBACH model instead of MODIS derived LAI does not significantly modify neither the spatial distribution nor the seasonal profile of the emissions. Sindelarova et al. (2014) however report a slightly different emission profile in comparison to this study and to Messina et al. (2015), with a shifted isoprene emission maximum around October/November and a minimum in June, and a maximum of monoterpenes emissions extended to July/August and beginning of September. The differences in the seasonal emission features are thus mainly due to the differences in the climatic forcing used in the three studies, climate variables (here temperature and radiation) being main drivers of the inter-annual fluctuations of emissions (Guenther et al., 2012).

## 4.3 Sensitivity simulations

This section presents the sensitivity of emission estimates with the biogenic module in ECHAM-HAMMOZ to a selection of model input parameters and parameterizations related to land and vegetation cover (emission factors, PFT distributions and





soil water). We also analyze the effect of nudging climate in different configurations of the biogenic module. Sensitivity of BVOC emissions to different LAI distributions has not been tested here separately, as the use of JSBACH derived LAI does not lead to significant changes in the emission distributions. LAI impact could be of importance if changing dynamically the vegetation cover in response to land-use and future climate changes. We only show here the impacts on the emissions of the

most abundant BVOCs. A detailed list of emission estimates for the 32 compounds simulated with the biogenic module is given in Supplementary Material, Sect. S4.

### 4.3.1 Impact of PFT-dependent emission factors

The impact of emission factors on the emissions calculated with the biogenic module in ECHAM-HAMMOZ are evaluated in simulation PFT-CLM4. Here, the emission factors from the global maps of 10 compounds (isoprene, $\alpha$-pinene, $\beta$-pinene, 3-

carene, limonene, myrcene, t-$\beta$-ocimene, sabinene, 232-MBO and nitric oxide) as used in the CTRL simulation are replaced by PFT-specific emission factors calculated from the MEGAN2.1-CLM4 PFT fractions (as explained in Sect. 2.3.2). Global and regional relative differences of the annual emissions of the 10 compounds listed above are shown in Fig. 5. Globally, isoprene emissions decrease by 8.5 % in experiment PFT-CLM4, resulting from a decrease of emission mainly in Australia, but also in Africa, Northern America and Russia. However, isoprene emissions increase in South America and South Asia. Sindelarova

et al. (2014) report a global decrease of 12.5 % for the same sensitivity experiment, and similar regional distribution of the impacts. The difference between the two studies can be attributed to the difference in the spatial resolution of the simulations (1.875° x 1.875° here and 0.5° x 0.5° in Sindelarova et al. (2014)), that significantly influences the emission estimates in coastal regions and areas with large variations in topography and land cover (Pugh et al., 2013). However, the impact of emission factors on isoprene emissions obtained here and by Sindelarova et al. (2014) is much higher than the 1 % decrease reported

in Guenther et al. (2012). Most of the differences obtained here can be explained directly from the differences in the emission factor distributions. The emission factor maps used in the reference simulation are derived from detailed land cover and plant species distributions, as well as above canopy flux measurements, and thus account for species composition variability Guenther et al. (2012). For the PFT-based emission factor distributions used in the PFT-CLM4 simulation, each PFT is associated to a constant value of emission factor per compound, which is an average of the plant species specific emission factors for the plant

species belonging to that particular PFT. For some PFTs, grouping together species with comparable emissions, the differences are small, but for PFTs grouping together high and low emitter plant species, this can lead to significant discrepancies. For example, Broadleaf Deciduous Temperate Trees PFT groups together *Acer* with negligible isoprene emissions and *Quercus* with high isoprene emissions. Switching to this PFT-specific emission factor thus lowers isoprene emission in the regions covered by *Quercus*. The biogenic emission module is highly sensitive to changes in emission factors. The modification of the

emission factor distribution mainly affects the spatial distribution of the simulated BVOC emissions, but does not impact the seasonality of the emissions.

As shown in Fig. 6, the increase of isoprene emission in tropical regions in the PFT-CLM4 simulation results directly from the higher constant values of emission factors associated to tropical PFTs. The decrease of isoprene emission in Australia and sub-Saharan Africa is directly linked to the covering of these regions with mainly temperate shrub and grass PFTs in the



MEGAN2.1-CLM4 PFT distribution, associated to lower constant emission factors. Moreover, the MEGAN2.1-CLM4 PFT distribution does not reproduce the presence of temperate and tropical species with higher emission factors, especially in the North of Australia. In the Northern part of North America and in Siberia, isoprene emission factors associated to temperate and boreal PFTs, dominant in these regions, are lower than the species-specific emission factors. However, despite the significant

lowering of emission factors in Northern regions, isoprene emission are only slightly decreased in comparison to tropical regions. This is mainly due to the lower rate of isoprene emissions in cooler climatic conditions (Guenther et al., 2006).

Monoterpenes emissions are strongly impacted by the use of PFT-specific emission factors. At the global scale, $\alpha$-pinene and myrcene emissions increase by respectively 75 % and 157 %, whereas t-$\beta$-ocimene emission decreases by 25 %. Large increases of all the monoterpene compounds occur in Australia in the PFT-CLM4 simulation, due to the presence in the

major part of Australia of temperate shrub PFTs, which are strong emitters of monoterpenes. Similarly to isoprene, $\alpha$-pinene and myrcene emissions increase strongly in South America (+125 % for $\alpha$-pinene and +300 % for myrcene) in response to the larger spatial coverage of tropical vegetation with high emission factors in the MEGAN2.1-CLM4 PFT distribution. In Northern regions, $\alpha$-pinene emissions also increase (+82 % in North America) in response to the high values of emission factors associated to needleleaf evergreen temperate and boreal PFTs, mainly covering these regions in the PFT-CLM4 simulation.

However, t-$\beta$-ocimene emissions decrease in all regions except Australia (for example, -34 % in South America, -39 % in Europe). This is due to the slightly lower emission factor associated to tropical and temperate PFTs for t-$\beta$-ocimene.

232-MBO and nitric oxid emissions are also significantly affected by the use of PFT-dependent emission factors. 232-MBO and nitric oxide emissions decrease by 18 % and 38 %. 232-MBO emissions increase strongly in Europe and Russia, but decrease in the rest of the world, whereas nitric oxid emissions decrease in all regions. As shown in Fig. 6, the decrease of

232-MBO emissions in tropical regions is due to the very low emission factor values associated to temperate and tropical PFTs (between 0.0002 and 0.04 mg C m$^{-2}$ day$^{-1}$) in comparison to the emission factor values deduced from global maps (between 0.1 and 0.5 mg C m$^{-2}$ day$^{-1}$). The much higher values of emission factors associated to needleleaf evergreen temperate and boreal tree PFTs (PFTs 1 and 2 in the MEGAN2.1-CLM4 PFT classification, with emission factors of respectively 14.4 and 1.23 mg C m$^{-2}$ day$^{-1}$) lead to a strong increase of 232-MBO emissions in northern regions and particularly in Northern Europe

and Russia mainly covered by PFTs 1 and 2. However, the PFT-based emissions factors are still lower than the species-specific values obtained from the global maps in North America, explaining the decrease in 232-MBO in several areas.

### 4.3.2 Impact of PFT distribution

In simulation PFT-JSBACH, the MEGAN2.1-CLM4 PFT distribution is replaced by the extended-JSBACH PFT distribution. As explained in Sect. 2.3.2, the extended-JSBACH 14-PFT classification has been developed to be as similar as possible to the

MEGAN2.1-CLM4 15-PFT classification, in order to use the emission factors associated to the MEGAN2.1-CLM4 PFTs for the extended-JSBACH PFTs. Therefore, differences in BVOC emissions between simulations PFT-JSBACH and PFT-CLM4 only results from the differences in the PFT geographical distribution and fractional coverage of the grid-cells. Table 6 lists the global continental areas covered by the 14 PFTs of the MEGAN2.1-CLM4 and extended-JSBACH classifications and Fig. 7 illustrates the distribution of dominant PFTs for both classifications. Tropical and temperate tree PFTs are generally less

extended in the JSBACH distribution. The areas covered by Broadleaf Deciduous Temperate and Tropical trees (PFT 2 and 7) are particularly reduced, whereas boreal tree PFTs cover larger areas. The areas covered by shrub PFTs are also lowered, especially for the boreal shrubs (PFT 10) which are replaced in the JSBACH distribution by cool/cold C3 grasses or bare soil at high latitudes of the Northern Hemisphere. Differences also occur in the grass and crop PFT distributions, but only slightly

impact BVOC emissions due to the lower emission factors associated to these PFTs.

At the global scale, all BVOC emissions decrease in the PFT-JSBACH experiment in comparison to the PFT-CLM4 experiment mainly in response to the reduction of the spatial extent of tropical PFTs (PFT 1 and 2). The seasonality of emission is not affected by the change in PFT distribution. Isoprene and monoterpenes emissions are reduced by about 9 %, methanol and acetone by 17 %, and beta-caryophyllene by 20 % (Fig. 8). The global reduction of isoprene emission due to PFT distribution

change we have obtained here is less than the 13 to 24 %, and 30 % impact reported respectively in Guenther et al. (2006) and Pfister et al. (2008). In these two studies, the PFT distributions used to force the MEGAN model were mainly obtained from MODIS satellite observations using different procedures to assign PFT cover. This caused large differences in PFT distributions and area estimates, and thus in isoprene emissions.

As shown in Fig. 9, the decrease of isoprene, $\alpha$-pinene (and other monoterpenes) and $\beta$-caryophyllene annual emissions

occurs mainly in tropical regions, due to the reduction of the spatial extension of tropical PFTs in the extended-JSBACH distribution, particularly in Indonesia and South Asia (see Fig. 10), which are strong emitters of most of the compounds. However, the extension of broadleaf deciduous tropical trees (PFT 2) in Central Africa, India and the South of China leads to higher emissions in these regions. In agreement with the results of Guenther et al. (2006) and Pfister et al. (2008), we observe here that isoprene emissions are mostly impacted by the differences in the coverage of broadleaf trees and shrubs, that are the

vegetation classes with the highest emission factors. Changes in crop, grass and needleleaf trees distributions have a smaller effect. Moreover, the effect of tropical PFT on the emissions are also reinforced by the year-long warm temperatures and high incoming radiation of tropical regions inducing higher activity factors, and thus higher emission rates. This effect is also illustrated by the relatively low impact of the modification in the boreal PFT distributions on isoprene emission. Indeed, despite the expansion at mid- and high latitudes of the Northern Hemisphere of Broadleaf Deciduous Boreal trees (PFT 8), associated

to high isoprene emission potential, isoprene emission are only slightly increased due to the lower values of activity factors for light and temperature in these regions.

232-MBO emissions are significantly decreased in Northern regions in response to the reduction of the fractions of Needleleaf Evergreen Temperate and Boreal trees (PFT 3 and 4). 232-MBO has the same parameterization for light and temperature as isoprene, but it is mainly emitted by boreal and cold tree species and its emission factor for Needleleaf Evergreen Temperate trees (PFT 3) is about 70,000 times its emission factor associated to tropical PFTs. $\alpha$-pinene and $\beta$-caryophyllene emissions

also decrease in Northern regions in response to the reduction of PFTs 3 and 4 coverage. These compounds don't have higher emission potentials for cold PFTs in comparison to tropical PFTs, but the different parameterization of the temperature activity factor, including a light-independent function, leads to higher emissions rates and emission sensitivity at high latitudes in comparison to isoprene.





### 4.3.3  Impact of soil moisture on isoprene emission

The impact of soil moisture on simulated isoprene emission is evaluated in experiment TEST-SM, via the activation of the activity factor $\gamma_{SM}$ as described by Guenther et al. (2006, 2012). The soil water activity factor is evaluated here using the relative soil water amount (soil water depth relative to the maximum water depth, corresponding here to the root zone (Hagemann and

Stacke, 2015)), and a wilting point set at 35 % of the maximum soil water amount. The detailed parameterization of the $\gamma_{SM}$ activity factor is given in Supplementary Material, Sect. S1.

The activation of the soil moisture activity factor reduces global isoprene emission by 15.2 %. The soil moisture impact reported here is within the range of previous estimates with the MEGAN model. It is higher than the 1 % and 7 % reductions in isoprene emission reported respectively by Lathière et al. (2010) and Guenther et al. (2006), but less than the 21 % reduction

of Müller et al. (2008) and 50% reduction of Sindelarova et al. (2014). All the mentioned studies are based on the same algorithm, thus the differences in the soil moisture effects on isoprene emissions are due only to the use of different soil moisture and wilting point datasets. Wilting point is particularly important since it is the threshold value below which the soil moisture activity factor, and thus the isoprene emissions are set to zero, and has to be consistent with the soil moisture data (Guenther et al., 2012). Nevertheless, the soil moisture impact has to be considered with caution (Müller et al., 2008), because

its parameterization is based on measurements from only one study (Pegoraro et al., 2004).

Despite the use of different soil moisture databases, the geographical distribution of the soil moisture activity factor showed in Fig. 11 is very similar to the ones obtained in Müller et al. (2008) and Sindelarova et al. (2014). Arid and semi-arid regions are characterized by $\gamma_{SM}$ values below 0.5, whereas most of the tropical and temperate regions have $\gamma_{SM}$ up to 0.9. Our estimates of $\gamma_{SM}$ are higher than the values reported by Sindelarova et al. (2014) in South Africa, Central Africa and South Asia, but

lower than the estimations of Müller et al. (2008) and Sindelarova et al. (2014) in Northern regions, especially in Siberia. Similarly to Müller et al. (2008) and Sindelarova et al. (2014), isoprene emissions mainly decrease in the tropical regions affected by low soil moisture activity factor, i. e. the Southern part of North America, South America, Subsaharan Africa, Central Asia and Australia. The decrease in isoprene emission is particularly marked in the North of Australia, reaching up to 60% of the reference emissions. The soil moisture impact is slightly more pronounced during summer months and dry seasons

(not showed), except in Australia where the major part of the annual effect is due to the emission reductions occurring during the Southern Hemisphere summer.

### 4.3.4  Impact of nudging

Since the soil moisture effect depends primarily on the modeled water cycle, we also tested the impact of running a nudged simulation versus simulations with only prescribed sea surface temperatures and sea ice coverage. Even though nudging in

ECHAM6-HAMMOZ does not include nudging to humidity data, the use of surface pressure, temperature, vorticity and divergence from reanalysis has an impact on the water cycle in ECHAM6 (Lohmann and Hoose, 2009; Lohmann and Ferrachat, 2010). The effect of nudging on the biogenic emissions is analyzed in experiments TEST-NUDG and TEST-NUDG+SM. Simulation TEST-NUDG is identical to the reference simulation except that nudging to ERA-Interim data is applied for vorticity,





divergence, temperature and surface pressure (Lohmann and Hoose, 2009; Zhang et al., 2014). In simulation TEST-NUDG+SM the nudging and the soil moisture activity factor are activated. Generally, nudging the climate simulation towards global weather reanalysis is applied in order to perform more straightforward comparison between simulation and observation, and also to allow for distinguishing between signal and noise after a shorter simulation time (Zhang et al., 2014). In the experiment TEST-

NUDG, nudging has a weak impact on the simulated global biogenic emissions, decreasing the total global emissions by 2.6 %. Figure 12 gives the annual global relative emission differences for the most emitted and important compounds. Isoprene global annual emissions are reduced by 1.8 % (-7.6 Tg C year$^{-1}$), which is less than the interannual variability (standard deviation of 9.1 Tg C year$^{-1}$ for the 13 years simulation), monoterpenes emissions by 3.6 %, and a maximum decrease of 5.2 % is obtained for $\beta$-caryophyllene. The activation of the soil moisture activity factor in the TEST-NUDG+SM simulation leads to similar

conclusions. The isoprene global emission is reduced by 17.1 % (relative to the reference simulation), which is about the sum of the soil moisture impact (-15.2 %) and the nudging impact (-1.8 %). By constraining the simulated wind and temperature fields using reanalysis, nudging reduces the BVOC emission variability in agreement with (Zhang et al., 2014). As illustrated in Fig. 13 for isoprene and monoterpenes, the interannual and seasonal variations of global emissions in the nudged and reference simulations have mostly similar profiles, but the annual and monthly fluctuations are reduced in the TEST-NUDG simulation

in comparison to the reference simulation by 1 Tg C year$^{-1}$ for isoprene and by 0.08 Tg C year$^{-1}$ for monoterpenes. The difference between the highest and the lowest global annual emission over the 2000-2012 simulation period is about 8.4 % (5.8 %) of isoprene (monoterpenes) total emission averaged over the simulation period in the reference simulation and 5.5 % (4.9 % for monoterpenes) in the nudged simulation. However, the activation of the soil moisture effect increases the interannual variability of isoprene emissions to 8.8 % in the TEST-NUDG+SM simulation. The temporal profile of the emission is similar

to the profile of experiment TEST-NUDG, but shifted to lower annual totals due to the reduction of isoprene emission by the soil moisture effect. The interannual variability for isoprene obtained here is lower than the 20 % and 29 % variability reported by respectively Müller et al. (2008) (1995-2006 MEGANv2 simulation forced with ECMWF reanalysis) and Sindelarova et al. (2014) (1980-2010 MEGANv2.1 simulation forced with MERRA reanalysis). However, the interannual variability of the reference simulation fairly agrees with the 8.5 % variability obtained by Lathière et al. (2006) for a 1983-1995 simulation using

the MEGANv2 model forced with satellite based climate archive. Differences in the isoprene variability can be associated to the differences in the meteorological datasets used in the different studies. The activation of the soil moisture activity factor could partly explain the larger interannual variability in the study of Müller et al. (2008), following the results of experiment TEST-NUDG+SM. The activation of the $CO_2$ activity factor in Sindelarova et al. (2014) can also lead to larger differences between the global isoprene estimates at the beginning and the end of the temporal series studied.

Nudging has also a weaker impact on the spatial distribution of BVOC emissions in comparison to PFT distributions and emission factors. Effects of nudging are mainly localized in tropical regions (see Fig. 14). Isoprene emissions increase in the South-East of North America, North of South America and the western part of Equatorial Africa, and slightly decrease in South America, India and large parts of Australia. The same distribution of emission differences is observed for $\alpha$-pinene, but the impacts are more marked in the Northern Hemisphere, with a decrease of emissions in North America, Southern Europe

and Russia, and an increase of emissions in Northern Europe. Isoprene and $\alpha$-pinene emission differences are mainly driven



by the differences in surface air temperature and shortwave radiation induced by the nudging of meteorological variables, as showed in Fig. 15. The increase of temperature and radiation in the North of South America, South-East of North America, Western Africa and Northern Europe leads to emission increases, whereas the decrease of temperature in Australia and North America induces emissions decrease. An exception is observed in India where the emission decreases is not directly linked to the temperature and radiation increase observed there. As discussed in Lohmann and Hoose (2009) and Lohmann and Ferrachat (2010), nudging the ECHAM model only weakly impact global mean TOA radiation and global temperature. Here, we obtain a global surface air temperature decrease of -0.17°C (most of the regional impacts counterbalancing each others) and a global shortwave surface radiation increase of 4.6 W/m$^2$. The decrease of emission in India could be indirectly linked to the large decrease of precipitation obtained there (see Fig. 15). Lohmann and Hoose (2009) already pointed out that nudging in ECHAM has a stronger effect on precipitation, generally increasing the convective activity in the tropical regions. The activation of the soil moisture activity factor in TEST-NUDG+SM affects the isoprene emission distribution exactly with the same patterns and intensity as in experiment TEST-SM (not showed). The decrease of isoprene emissions is slightly more marked notably in India and in South America, due to the reduction of soil moisture in these regions in response to the large reduction of precipitation obtained in the nudged experiments.

## 5 Conclusions

A biogenic emission scheme based on the MEGANv2.1 model (Guenther et al., 2012) has been integrated into ECHAM6-HAMMOZ chemistry climate model and linked to parameters depending on the vegetation model JSBACH, in order to calculate the biogenic emissions of 32 compounds. The model calculates a total emission of 634 Tg C yr$^{-1}$ from terrestrial vegetation over the period 2000-2012 . Isoprene is the main contributor to the average emission total accounting for 66 % (417 Tg C yr$^{-1}$), followed by several monoterpenes (12 %), methanol (7 %), acetone (3.6 %) and ethene (3.6 %). Tropical regions are identified as the primary source of global BVOCs, contributing to 87 % of isoprene, 79 % of monoterpenes and 73 % of methanol global annual emissions. The interannual variability in biogenic emissions over the studied period is 8.4 %, slightly lower than previously reported (Sindelarova et al., 2014; Müller et al., 2008). The biogenic emission estimates in ECHAM6-HAMMOZ are within the range of previous emission budgets obtained with different versions of the MEGAN model. Nevertheless, model estimates of BVOC emissions show a large variation, global isoprene emissions varying within a factor of 1.6, when global monoterpenes and methanol emissions vary within a factor of about 3.5. Most of the discrepancies between model emission estimates can be attributed the use of different climatic input data, vegetation related parameters (LAI, PFT distributions), parameterizations and empirical coefficients within the MEGAN model (emission factors, parameterization of activity factors,...). The variability in BVOC emission estimates using the same model with different configurations highlights the need for a systematic effort to improve the understanding of the key processes and mechanisms responsible for BVOC emissions. Measurements of BVOC emissions in various regions of Earth and over longer time period, as well as sensitivity tests and model intercomparisons, are therefore required to obtain more accurate emission estimates and refine the model parameterizations.





The sensitivity of biogenic emissions to a selection of model input parameters and parameterizations related to vegetation cover and climate have been evaluated in a series of sensitivity simulations. The biogenic model shows a high sensitivity to the changes in PFT distributions and associated emission factors. The use of emission factors derived from PFT distributions instead of gridded maps of species-specific values results in isoprene and $\alpha$-pinene estimates varying respectively by 8.5 % and

75 % of the reference simulation values, and to the largest changes in the spatial distribution of BVOC emissions. These effects are mainly explained by the differences in the PFT spatial coverage and the averaged emission factors associated to each PFT in comparison to the species-specific values. Switching to the PFT distribution derived from the JSBACH vegetation model has a lower impact on BVOC emissions, and causes a decrease of isoprene and monoterpenes emissions by about 9 %, that can be mostly attributed to the differences in the distribution of tropical and temperate tree PFTs. Isoprene emissions show the highest

sensitivity to soil moisture impact, with a global decrease of isoprene emission by 12.5 % when the soil moisture activity factor is included in the emission parameterization. This effect is within the broad range of previous results of sensitivity studies, varying from 1 % (Lathière et al., 2010) to 50 % (Sindelarova et al., 2014). The large uncertainties concerning the soil moisture impact on isoprene emissions are mainly explained by the use of different soil moisture and wilting point databases. This also highlight the need of a better understanding and more constrained parameterization of the soil moisture impact on isoprene and

the introduction of it's impact on other compounds in the MEGAN model as suggested by Wu et al. (2015). Nudging ECHAM6 climate towards ERA-Interim reanalysis has a much lower impact on the biogenic emissions in comparison to the effects of PFT distributions and soil water. Constraining meteorological variables lowers the interannual variability in comparison to the reference simulation, and most of the regional impacts can be explained by the slight differences obtained in temperature and radiation.

The results of the present study demonstrate the capability of the biogenic model embedded in ECHAM6-HAMMOZ in reasonably representing BVOC emissions at the global and regional scales. This version of the ECHAM6-HAMMOZ model is now suitable for many tropospheric investigations, notably concerning the impact of BVOC emissions on the ozone budget, secondary aerosol formation and atmospheric chemistry. Activating the dynamic vegetation component of the JSBACH model also allows for the study of present-day and future land cover and land-use change impacts on atmospheric chemistry in a

comprehensive chemistry climate model framework.

## 6   Code availability

The code of the biogenic module (Fortran 95) is available upon request from the corresponding author or as part of the ECHAM6-HAMMOZ chemistry climate model through the HAMMOZ distribution web page https://redmine.hammoz.ethz.ch/projects/hammoz.

*Author contributions.* A.-J. Henrot implemented the biogenic module into the atmospheric chemistry model and performed the simulations. T. Stanelle and C. Siegenthaler worked on the link between the biogenic module and the vegetation model JSBACH. D. Taraborrelli and S.



Schröder helped in model developments. M. G. Schultz supervised the whole work and especially the design of the experiments. A.-J. Henrot prepared the manuscript with contributions from all co-authors.

*Competing interests.* The authors declare that they have no conflict of interest.

*Acknowledgements.* We are grateful to Christian Reick for discussions on model developments and results. This research is based upon work
5  co-funded by the European Union (BeIPD- Marie Curie COFUND). The ECHAM-HAMMOZ model is developed by a consortium composed of ETH Zurich, Max Planck Institut für Meteorologie, Forschungszentrum Jülich, University of Oxford, the Finnish Meteorological Institute and the Leibniz Institute for Tropospheric Research, and managed by the Center for Climate Systems Modeling (C2SM) at ETH Zurich.



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



**Table 1.** Biogenic compound classes and individual compounds included in the biogenic emission module.

| Compound Classes | Compound Names |
| --- | --- |
| isoprene | isoprene |
| myrcene | myrcene |
| sabinene | sabinene |
| limonene | limonene |
| 3-carene | 3-carene |
| t-$\beta$-ocimene | t-$\beta$-ocimene |
| $\alpha$-pinene | $\alpha$-pinene |
| $\beta$-pinene | $\beta$-pinene |
| $\beta$-caryophyllene | $\beta$-caryophyllene |
| 232-MBO | 2-methyl-3-buten-2-ol |
| methanol | methanol |
| acetone | acetone |
| carbon monoxide | carbon monoxide |
| nitric oxide | nitric oxide |
| bidirectional VOCs | ethanol |
|  | acetaldehyde |
|  | formaldehyde |
|  | acetic acid |
|  | formic acid |
| stress VOCs | ethene |
|  | hydrogen cyanide |
|  | toluene |
| other VOCs | methyl bromide |
|  | methyl chloride |
|  | methyl ioide |
|  | dimethyl sulfide |
|  | methane |
|  | ethane |
|  | propane |
|  | butene |
|  | propene |
|  | benzaldehyde |





**Table 2.** Original (11-PFT) and extended (14 PFT) JSBACH PFT classifications for use in the biogenic emission module and correspondence to the MEGAN2.1-CLM4 15-PFT classification (as listed in Guenther et al. (2012)).

| JSBACH 11 PFTs | JSBACH extended 14 PFTs (and *corresponding MEGAN2.1-CLM4 PFT number*) | |
|---|---|---|
| (1) Tropical Evergreen Trees | (1) Broadleaf Evergreen Tropical Trees *(4)* | |
| (2) Tropical Deciduous Trees | (2) Broadleaf Deciduous Tropical Trees *(6)* | |
| (3) Extra Tropical Evergreen Trees { | (3) Needleleaf Evergreen Temperate Trees | *(1)* |
| | (4) Needleleaf Evergreen Boreal Trees | *(2)* |
| | (5) Broadleaf Evegreen Temperate Trees | *(5)* |
| (4) Extra Tropical Deciduous Trees { | (6) Needleleaf Deciduous Boreal Trees | *(3)* |
| | (7) Broadleaf Deciduous Temperate Trees | *(7)* |
| | (8) Broadleaf Deciduous Boreal Trees | *(8)* |
| (5) Raingreen Shrubs | (9) Broadleaf Deciduous Temperate Shrubs *(10)* | |
| (6) Deciduous Shrubs | (10) Broadleaf Deciduous Boreal Shrubs *(11)* | |
| (7) C3 Grasses { | (11) Cold/arctic C3 Grasses | *(12)* |
| | (12) Cool C3 Grasses | *(13)* |
| (9) C3 Pastures | (12) Cool C3 Grasses *(13)* | |
| (8) C4 Grasses | (13) Warm C4 Grasses *(14)* | |
| (10) C4 Pasture | (13) Warm C4 Grasses *(14)* | |
| (11) C3 and C4 Crops | (14) Crops *(15)* | |





**Table 3.** List of simulations performed.

| Experiment | Description |
| --- | --- |
| CTRL | reference simulation |
| PFT-CLM4 | use of MEGAN2.1-CLM4 PFT-specific emission factors |
| PFT-JSBACH | use of JSBACH PFT-specific emission factors |
| TEST-SM | impact of soil moisture on isoprene ($\gamma_{SM}$) |
| TEST-NUDG | impact of climate nudging |
| TEST-NUDG+SM | impact of climate nudging and soil moisture on isoprene |

**Table 4.** Global annual emission totals averaged over the 12 years (2000-2012) of the reference simulation for the 32 compounds and relative contributions to the global annual emission total. Averaged values are given with the standard deviation $\sigma$.

| Compound Names | Global annual emission (Tg (species) yr$^{-1}$) | Global annual emission (Tg C yr$^{-1}$) | Relative contribution (%) |
|---|---|---|---|
| isoprene | 473±10.2 | 417±9.1 | 65.8 |
| methanol | 117.1±1.6 | 43.9±0.6 | 6.9 |
| $\alpha$-pinene | 29.6±0.5 | 26.1±0.4 | 4.1 |
| acetone | 36.9±0.5 | 22.9±0.3 | 3.6 |
| ethene | 26.8±0.4 | 22.9±0.4 | 3.6 |
| $\beta$-pinene | 17.9±0.3 | 15.8±0.2 | 2.5 |
| t-$\beta$-ocimene | 15.9±0.3 | 14±0.2 | 2.2 |
| propene | 14.8±0.2 | 12.7±0.2 | 2.0 |
| ethanol | 16.8±0.3 | 8.8±0.2 | 1.4 |
| acetaldehyde | 16.8±0.3 | 9.2±0.2 | 1.4 |
| limonene | 9.5±0.1 | 8.4±0.1 | 1.3 |
| butene | 7.4±0.09 | 6.3±0.08 | 1.0 |
| 3-carene | 6.8±0.1 | 6±0.1 | 1.0 |
| sabinene | 6.4±0.1 | 5.7±0.1 | 0.9 |
| $\beta$-caryophyllene | 4.8±0.1 | 4.3±0.1 | 0.7 |
| myrcene | 2.6±0.04 | 2.3±0.04 | 0.4 |
| 232-MBO | 2.4±0.1 | 2±0.1 | 0.3 |
| formaldehyde | 4.2±0.07 | 1.7±0.03 | 0.3 |
| acetic acid | 3.1±0.06 | 1.3±0.02 | 0.2 |
| toluene | 1.4±0.02 | 1.3±0.02 | 0.2 |
| formic acid | 3.1±0.06 | 0.8±0.02 | 0.1 |
| hydrogen cyanide | 0.7±0.01 | 0.3±0.005 | 0.05 |
| ethane | 0.3±0.004 | 0.3±0.003 | 0.04 |
| methane | 0.15±0.002 | 0.12±0.001 | 0.02 |
| methyl chloride | 0.3±0.004 | 0.07±0.0009 | 0.01 |
| dimethyl sulfide | 0.09±0.001 | 0.04±0.0005 | 0.006 |
| propane | 0.03±0.0004 | 0.03±0.0003 | 0.004 |
| benzaldehyde | 0.03±0.0004 | 0.02±0.0003 | 0.004 |
| methyl bromide | 0.06±0.0008 | 0.01±0.0001 | 0.001 |
| methyl ioide | 0.03±0.0004 | 0.003±0.0001 | 0.0004 |
| carbon monoxide | 95.8±1.5 | 41.1±0.6 | - |
| nitric oxide | 5.9±0.07 | - | - |
| Total | | 634.1±12.5 | |



**Table 5.** Comparison of several BVOC global total emissions (Tg C year$^{-1}$) to previous published global totals with different versions of the MEGAN model.

| Ref. | Isoprene | Monoterpenes | Methanol | Acetone | Ethene | $\beta$-caryophyllene | Acetaldehyde | Formaldehyde | Acetic Acid | Formic Acid | 232-MBO | CO |
|---|---|---|---|---|---|---|---|---|---|---|---|---|
| (-) | 417 | 78.3 | 43.9 | 22.9 | 22.9 | 4.3 | 9.2 | 1.7 | 1.3 | 0.8 | 2 | 41.1 |
| (a) | 427.6 | 74.4 | 40.9 | 20.5 | | | 8.7 | 1.6 | 1.2 | 0.8 | 1 | |
| (b) | 464.6 | 91.3 | 37.8 | 24.6 | | | 8.6 | 1.9 | 1.1 | 0.7 | 1.3 | |
| (c) | 523.7 | 83.7 | 48.7 | 23 | 15.5 | | 10.4 | 1.8 | 1.4 | 0.9 | 1.4 | 38.6 |
| (d) | 471.6 | 124 | 37.3 | 24.8 | 23 | 6.5 | 11.4 | 2 | 1.5 | 1 | 1.9 | 35 |
| (e) | | | | 19.8 | | | | | | | | |
| (f) | 378 | | | | | | | | | | | |
| (g) | 393.2 | 78.5 | | | | | | | | | | |
| (h) | | | 39.3 | | | | | | | | | |
| (i) | 414 | 80 | | | | | | | | | | |
| (j) | 413 | | | | | | | | | | | |
| (k) | | | | | | | 12.5 | | | | | |
| (l) | 523 | | | | | | | | | | | |
| (m) | 469.9 | | | | | | | | | | | |
| (n) | 361.4 | | | | | | | | | | | |
| (o) | | | 30 | | | | | | | | | |
| (p) | 529 | | | | | | | | | | | |
| (q) | 460 | 117 | 106 | 42 | | | 15 | 10 | 0.3 | 1.5 | | |
| (r) | 601 | 103 | | | | | | | | | | |
| (s) | | | 48 | | | | | | | | | |
| (t) | 507 | 33 | | | | | | | | | | |
| (u) | | | | | 21.7 | | | | | | | |
| (v) | 503 | 127 | | | | | | | | | | |

(-) This study; (a) MEGANv2.1 off-line, 2000-2009 (Messina et al., 2015); (b) MEGANv2.1 in ORCHIDEE, 2000-2009 (Messina et al., 2015); (c) MEGANv2.1 off-line, 1980-2010 (Sindelarova et al., 2014); (d) MEGANv2.1 in CLM, 2000 (Guenther et al., 2012); (e) MEGANv2 in GEOS-Chem , 2006 (Fischer et al., 2012); (f) MEGANv2 off-line, 1981-2002 (Arneth et al., 2011); (g) MEGANv2 in ECHAM5-HAM, 2000 (O'Donnell et al., 2011); (h) MEGANv2.1 with MOHYCAN canopy (Stavrakou et al., 2011); (i) MEGANv2 in MOZART4, 2000-2007 (Emmons et al., 2010); (j) MEGAN in SDGVM (Lathière et al., 2010); (k) MEGANv2.1 in GEOS-Chem, 2004 (Millet et al., 2010); (l) MEGANv2 in CCSM3, 2000 (Heald et al., 2009); (m) MEGANv2 in MOZART4, 2005 (Pfister et al., 2008); (n) MEGANv2 with MOHYCAN canopy, 1995-2006 (Müller et al., 2008); (o) MEGANv2 in GEOS-Chem, 2004 (Millet et al., 2008); (p) MEGANv2 off-line, 2003 (Guenther et al., 2006); (q) MEGAN in ORCHIDEE, 1983-1995 (Lathière et al., 2006); (r) MEGAN in ISAM, 1981-2000 (Tao and Jain, 2005); (s) MEGAN in GEOS-Chem, 2001 (Jacob et al., 2005); (t) MEGAN in CLM, 1990 (Levis et al., 2003); (u) MEGAN in GEOS-Chem, 1993-1994 (Jacob et al., 2002); (v) MEGAN, 1990 (Guenther et al., 1995)



**Table 6.** Continental surface areas covered by the MEGAN2.1-CLM4 PFTs and the extended-JSBACH PFTs (m$^{12}$).

| PFT number and name | MEGAN2.1-CLM4 PFT surface | extended-JSBACH PFT surface |
| --- | --- | --- |
| 0 Bare soil | 55.4 | 85.8 |
| 1 Broadleaf Evergreen Tropical Trees | 14.1 | 11.75 |
| 2 Broadleaf Deciduous Tropical Trees | 8.19 | 2.98 |
| 3 Needleleaf Evergreen Temperate Trees | 4.77 | 1.52 |
| 4 Needleleaf Evergreen Boreal Trees | 10.68 | 8.22 |
| 5 Broadleaf Evergreen Temperate Trees | 2.18 | 1.25 |
| 6 Needleleaf Deciduous Boreal Trees | 1.49 | 2.24 |
| 7 Broadleaf Deciduous Temperate Trees | 5.32 | 1.81 |
| 8 Broadleaf Deciduous Boreal Trees | 1.98 | 3.19 |
| 9 Broadleaf Deciduous Temperate Shrubs | 6.32 | 3.62 |
| 10 Broadleaf Deciduous Boreal Shrubs | 8.97 | 1.15 |
| 11 Cold/arctic C3 Grasses | 4.66 | 4.32 |
| 12 Cool C3 Grasses | 13.56 | 11.54 |
| 13 Warm C4 Grasses | 12.46 | 16.25 |
| 14 Crops | 13.96 | 10.45 |





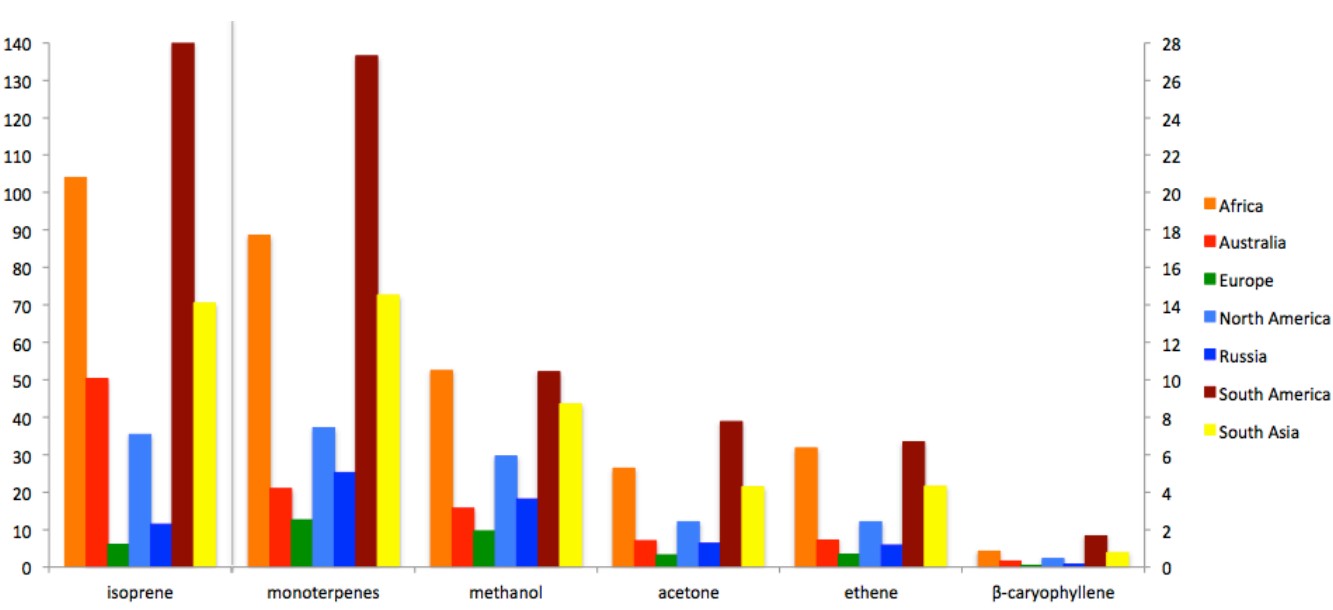

**Figure 1.** Regional annual emissions (Tg C yr$^{-1}$) for isoprene, the sum of monoterpenes, methanol, acetone, ethene and $\beta$-caryophyllene averaged over the 12 years (2000-2012) of the reference simulation. y-axes scales are separated for isoprene and the other compounds.





**Figure 2.** Global, Northern and Southern Hemispheres monthly emissions (Tg C month$^{-1}$) for isoprene, the sum of monoterpenes, methanol, acetone, ethene and $\beta$-caryophyllene averaged over the 12 years (2000-2012) of the reference simulation.





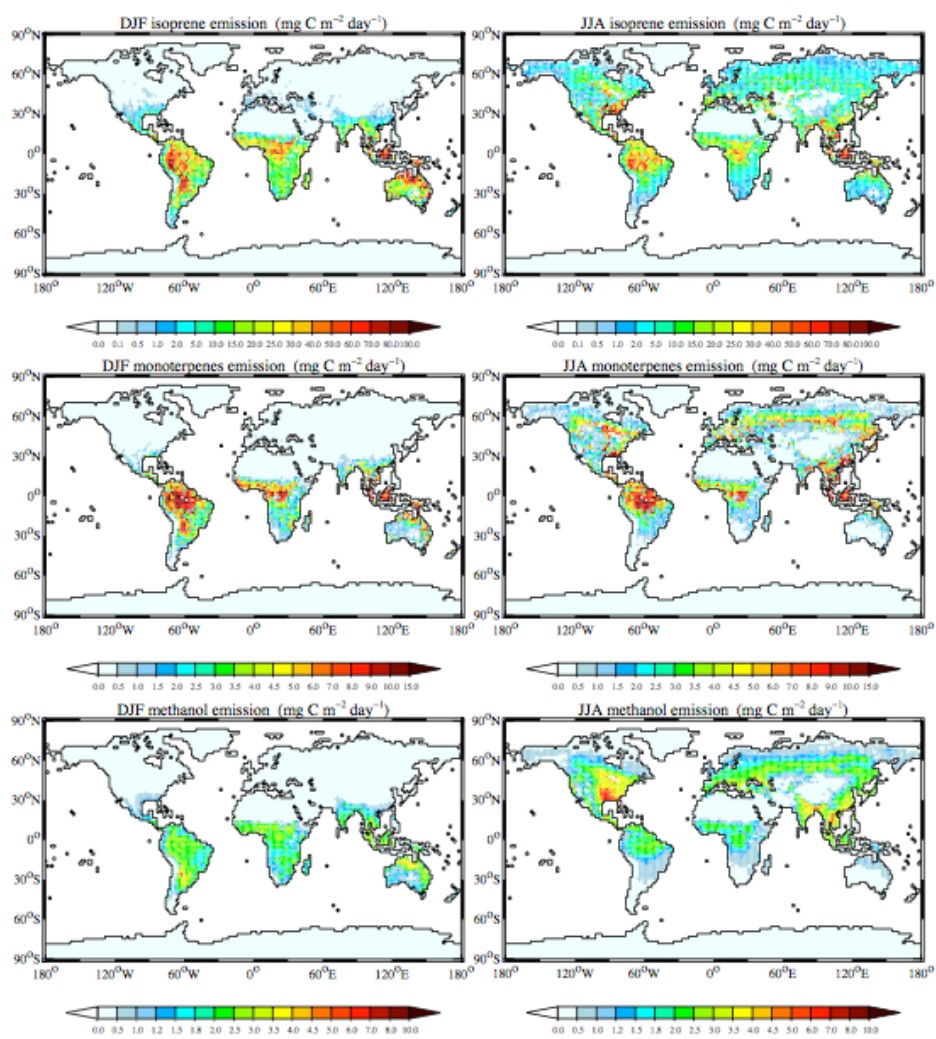

**Figure 3.** Spatial distribution of global December-January-February (DJF) and June-July-August (JJA) emissions (mg C m$^{-2}$ day$^{-1}$) for isoprene, the sum of monoterpenes and methanol averaged over the 12 years (2000-2012) of the reference simulation.



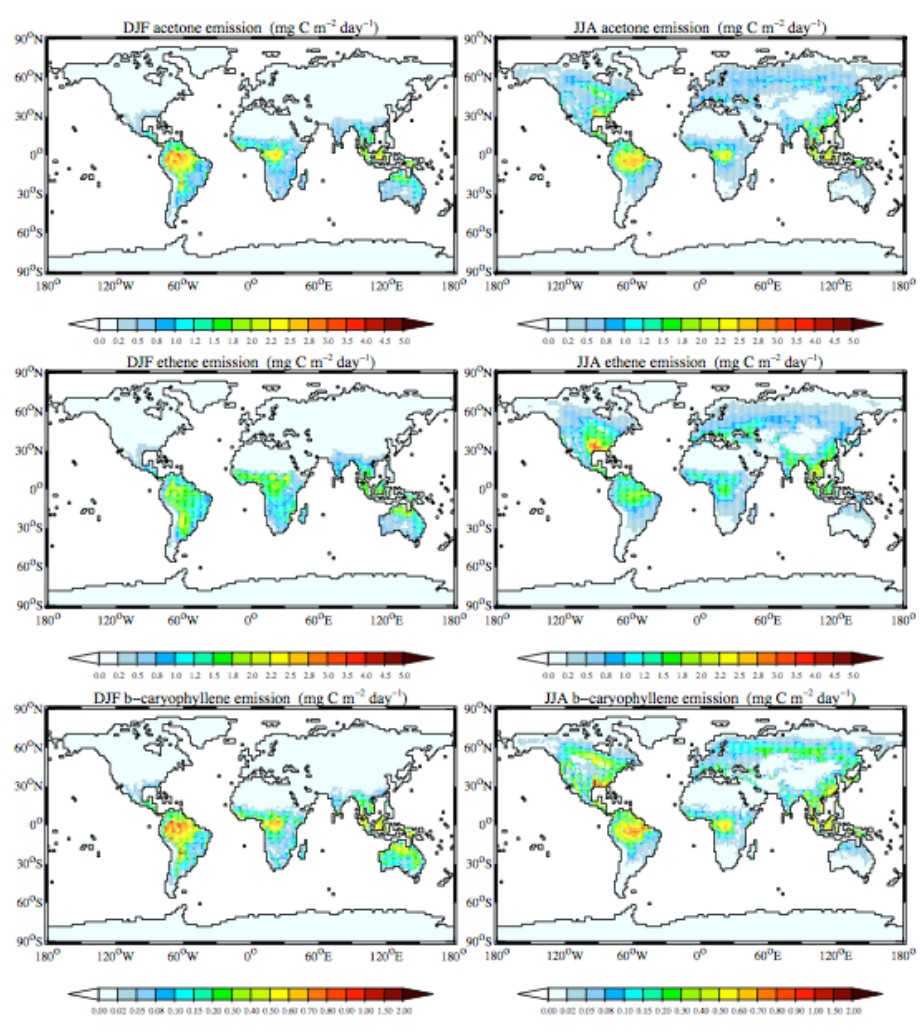

**Figure 4.** Same as Fig. 3 for acetone, ethene and $\beta$-caryophyllene.





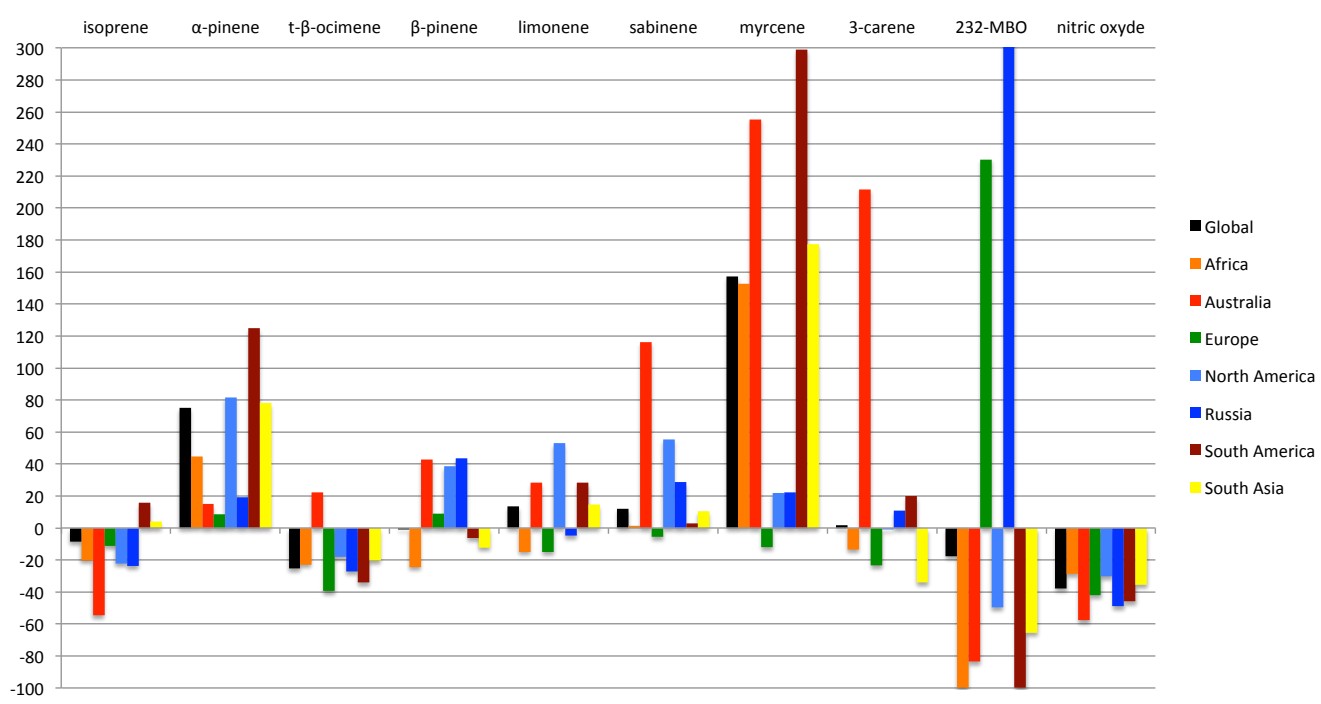

**Figure 5.** Global and regional relative emission differences (%) between the sensitivity simulation (PFT-CLM4) and the reference simulation for isoprene, monoterpenes, 232-MBO and nitric oxide.





**Figure 6.** a) Difference in isoprene annual emissions (mg C m$^{-2}$ day$^{-1}$) between the PFT-CLM4 and reference simulation, b) spatial distribution of emission factors (mg C m$^{-2}$ day$^{-1}$) derived from global maps for isoprene , c) spatial distribution of emission factors (mg C m$^{-2}$ day$^{-1}$) derived from MEGAN-CLM4 PFT distributions for isoprene; d), e), f) same as a), b), c) for $\alpha$-pinene; g), h), i) same as a), b),c) for 232-MBO.





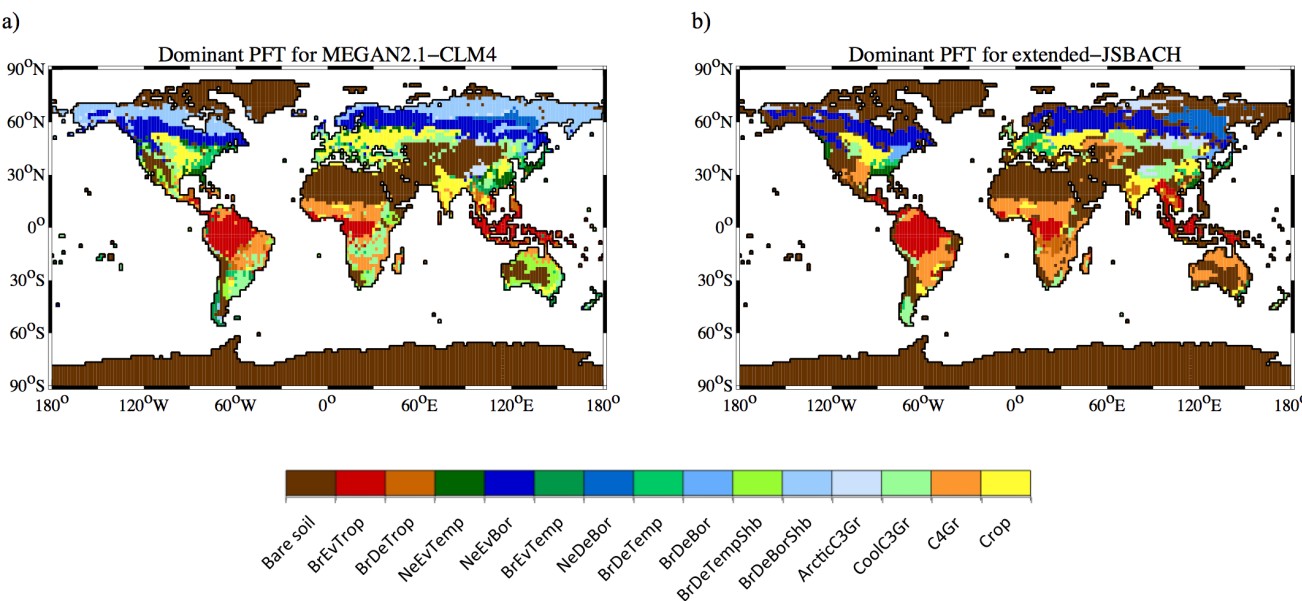

**Figure 7.** Distribution of dominant PFTs for a) MEGAN2.1-CLM4 and b) extended-JSBACH classifications.





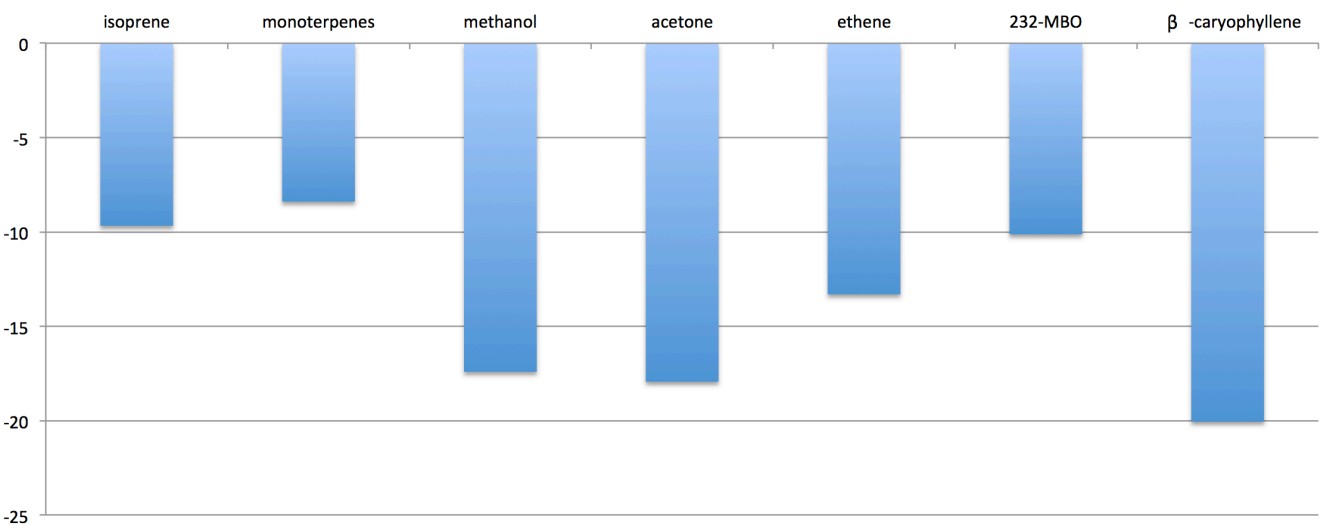

**Figure 8.** Global relative emission differences (%) between the sensitivity simulations PFT-JSBACH and PFT-CLM4 for isoprene, monoter-
penes, methanol, acetone, ethene, 232-MBO and $\beta$-caryophyllene.





**Figure 9.** Difference in annual emissions (mg C m$^{-2}$ day$^{-1}$) between the PFT-JSBACH ad PFT-CLM4 simulations for a) isoprene, b) $\alpha$-pinene, c) 232-MBO, and d) $\beta$-caryophyllene.





**Figure 10.** Difference between the extended-JSBACH PFT and the corresponding MEGAN2.1-CLM4 PFT distributions (cover fraction in %) for a) Broadleaf Evergreen Tropical trees, b) Broadleaf Deciduous Tropical trees, c) Needleleaf Evergreen Boreal Trees, and d) Broadleaf Deciduous Boreal Trees.





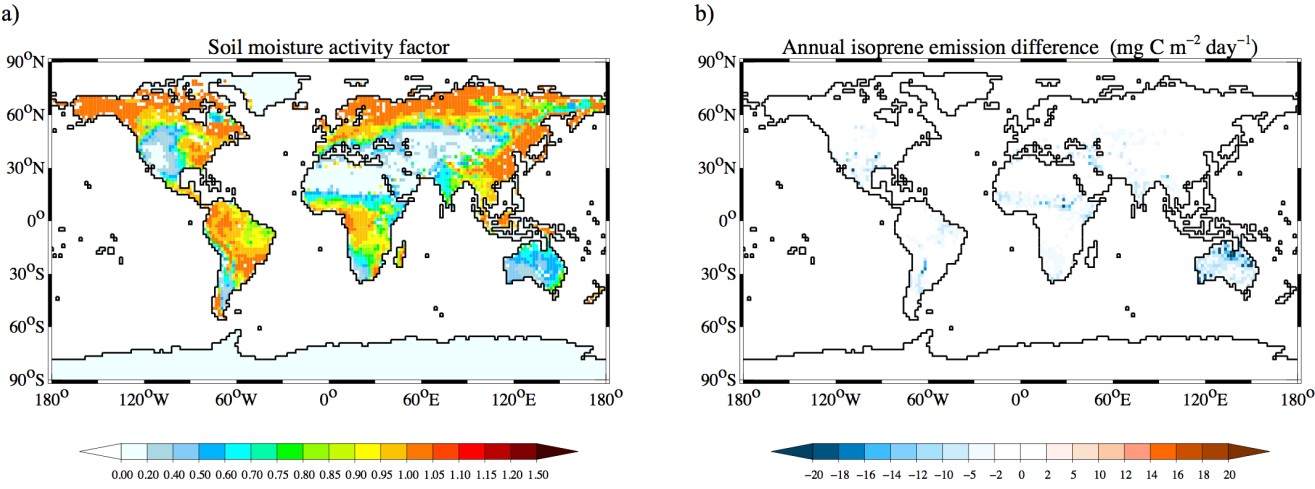

**Figure 11.** a) Annual soil moisture activity factor averaged over the simulation period, and b) annual isoprene emission differences (TEST-SM minus reference simulation) (Tg C year$^{-1}$).





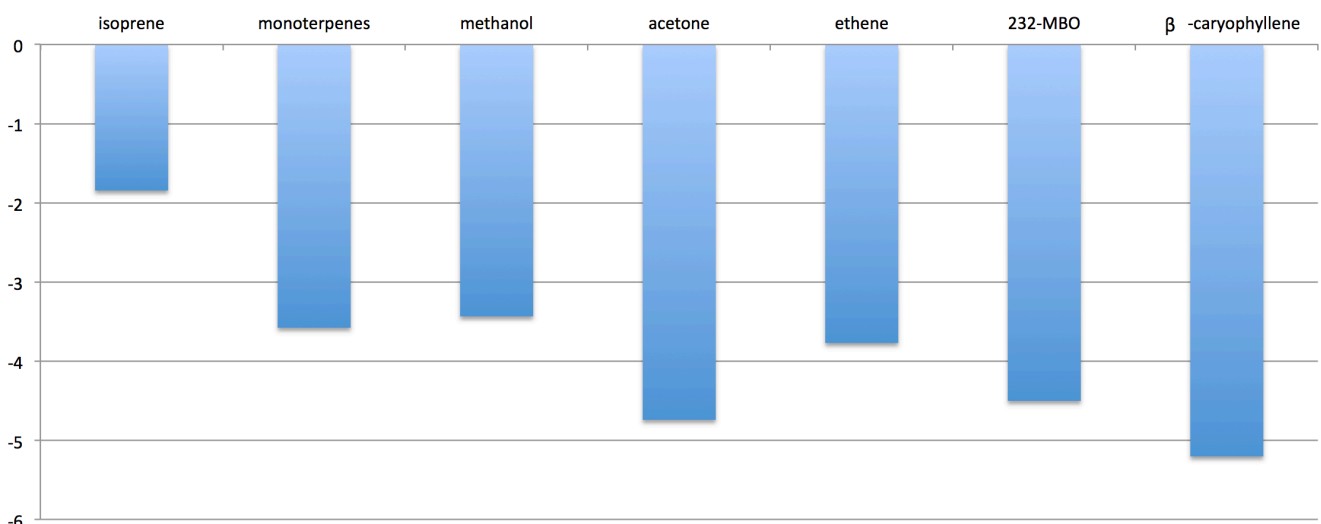

**Figure 12.** Global relative emission differences (%) between the sensitivity simulations TEST-NUDG and the reference simulation for isoprene, monoterpenes, methanol, acetone, ethene, 232-MBO and $\beta$-caryophyllene.





**Figure 13.** Temporal profiles of isoprene and monoterpenes global annual emissions (Tg C year$^{-1}$) (upper panel) and global monthly emissions (Tg C month$^{-1}$) (bottom panel) for the simulated 2000-2012 period.





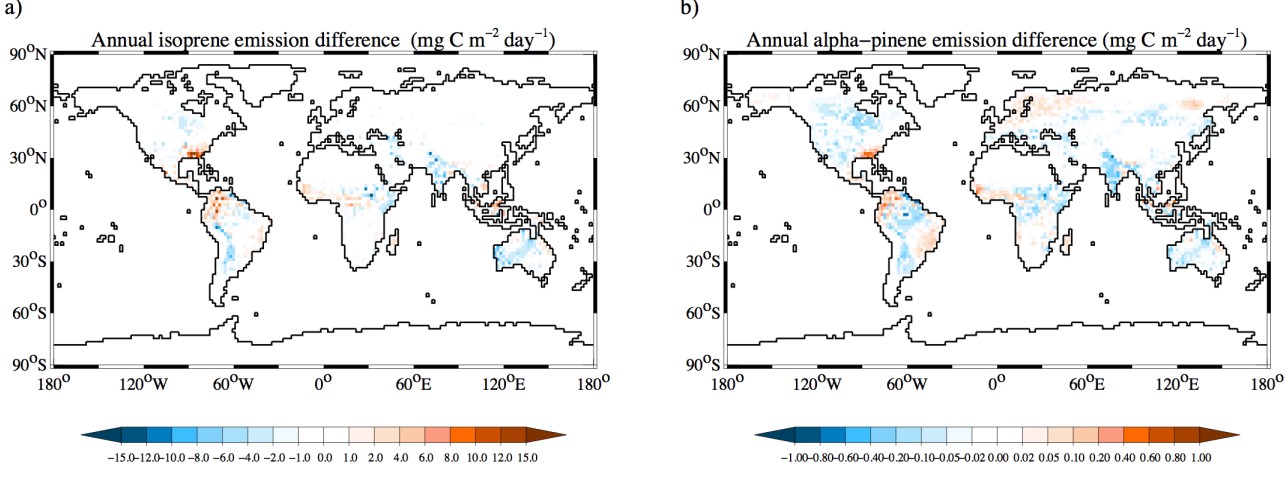

**Figure 14.** a) annual isoprene emission differences (TEST-NUDG minus reference simulation), and b) annual $\alpha$-pinene emission differences (TEST-NUDG minus reference simulation).





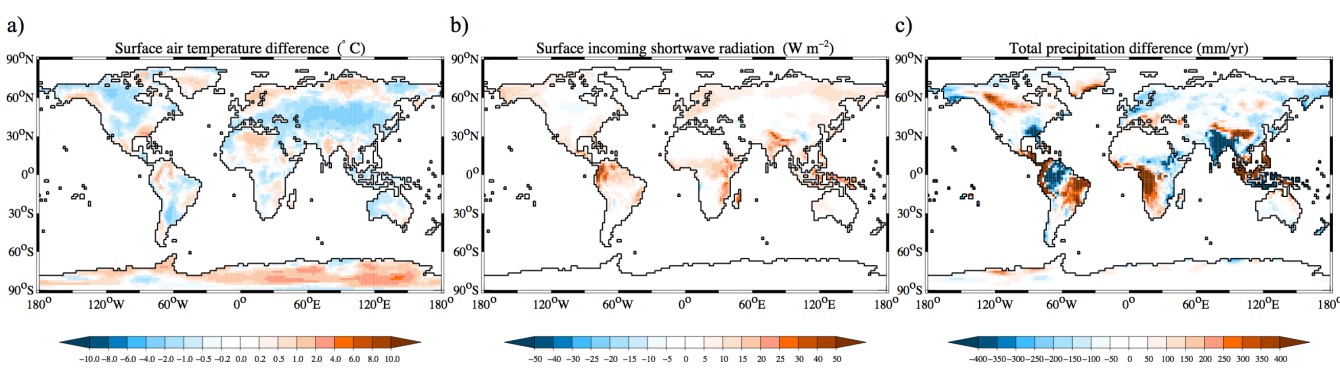

**Figure 15.** a) annual surface air temperature, b) annual surface incoming shortwave radiation, and c) annual total precipitation differences (TEST-NUDG minus reference simulation).