# Peer review of "Implementation of the biogenic emission model MEGAN(v2.1) into the ECHAM6-HAMMOZ chemistry climate model."

_Geoscientific Model Development, 2016_

## Referee Comment (RC1) · Anonymous Referee #1 · 27 Oct 2016

**Major comment**

This is a well-written and useful description of the implementation of MEGAN into a chemistry climate model, providing also interesting comparisons with a large number of previous MEGAN-based modelling studies. In addition, sensitivity simulations were performed to quantify the impact of different factors influencing the estimated emissions. The overall results appear very reasonable and are pretty well described and diagnosed.

My only major objection lies is the use of a hybrid MEGAN model algorithm which

mixes elements of the version 2.1 (Guenther et al., 2012) with the PCEEA developed for isoprene as part of MEGAN v2 (Guenther et al., 2006) and with the earlier algorithm of Guenther et al. (1993). The reader might want to know whether this setup provides results similar to those of the full version 2.1. Besides problems with the temperature dependence of light-independent emissions (see Specific comments, below), I am especially worried that the dependence on LAI is not correct. In Equation 3, I assume that $\gamma_{LAI}$ multiplies both ligh-dependent and light-independent terms. The expression of $\gamma_{LAI}$ is obtained from PCEEA, which is fine for light-dependent emissions, but certainly not for light-independent emissions, which are essentially proportional to LAI (cf. Equation 2 in Guenther et al., 2012). It is possible, although I'm not certain, that the authors were misled by a recently published model study (Messina et al., ACPD 15, 33967-34033, 2015). Messina et al. made the very surprising claim that, according to MEGAN, monoterpenes emissions show very little sensitivity (less than isoprene emissions) to changes in LAI. This result cannot be correct, and is contradicted by their reported results obtained with the ORCHIDEE model. Basically, the emission is proportional to the amount of leaf biomass, which is proportional to LAI. For light-dependent emissions, light attenuation dampens this relation. For light-independent emissions, this effect does not exist. Could the authors also estimate the sensitivity of their estimated isoprene and monoterpenes emissions to a given change in LAI (say, a factor of 1.5)?

To summarize, I recommend this article for publication in GMD, if the authors address this major objection, as well as the minor comments listed below.

**Specific comments**

The title is long, I think that its second part (Basic results and sensitivity test) could be dropped.

Page 3 line 30: please provide some more details on those global potential land cover maps. In what sense are those for potential land cover? Do they include realistic

representation of human influence (e.g. crops, managed forests, etc.)?

Page 4 line 25: the common MEGAN assumption that LDF is a species-specific constant is very unrealistic for monoterpenes (e.g. Rinne, Atmos. Chem. Phys. Discuss., 15, C11977–C11979, 2016). This should be (at least) mentioned somewhere in the manuscript.

Page 4 line 26: the temperature activity factor for light-dependent emissions is obtained from G06 (Guenther et al., 2006) but I find several differences between G06 and the expressions given in the supplement, e.g. the factor $C_{eo}$ is equal to 1.75 for isoprene in G06. Please explain.

Page 4 line 28: this temperature dependence (Guenther et al. 1993) is considerably simpler than the temperature activity factor of Guenther et al. 2012. Please discuss the possible implications of this simplification.

Page 7 line 14: the globemission website does not provide the box locations. The latitudes/longitudes of the regions should be specified in the figure legend.

Figures 3, 4, 6, 15: please enlarge the fonts, or enhance resolution for better readability

Page 8, lines 30-32: Messina et al. fail to provide good reasons for this supposed lower sensitivity of BVOC emissions to LAI in MEGANv2.1. The ORCHIDEE sensitivity to LAI makes much more sense.

Page 13 lines 2-3: here the soil water activity factor is said to depend on relative soil water amount. But the supplement reports a dependence on volumetric water content, as in G06. Are those two quantities the same thing?

Page 14 lines 21-24: Interannual variability is not well quantified by the ratio of maximum and minimum values. Since the periods covered by the different studies are all different, I'm afraid that the comparison amounts to a comparison of apples with oranges. It would make more sense to compare the standard deviation of annual totals in the different datasets.

Page 15 lines 8-9: I suppose that the process working on here is the influence of precipitation on soil moisture and hence on $\gamma_{SM}$. If so, this should be stated explicitly.

**Technical comments**

Abstract, line 4: please specify that the total given (634 Tg C year-1) is for the reference simulation.

Abstract, line 10: replace "with the biogenic model" by "here"

Abstract line 14: "PFT-dependent emission factor distribution" is unclear, use rephrase.

Page 2 line 5: replace "as well" by "also"

Page 2 lines 24-25: each reference is given twice, delete the second occurrence

Page 3 line 4: remove comme after "reactions"

Page 3 line 15: insert degree sign (°) after 1.8

Page 4 line 9: "emission potential (...) into the canopy" is weird. The BVOCs are emitted BY the canopy.

Page 4 line 10: remove space before "i. e."

Page 4 equation (3): a parenthesis is missing in this equation.

Page 4 lines 24-25: "The activity factor for temperature is divided into (...) factor" is unclear and could be rephrased e.g. as "Different expressions for the activity factor for temperature are considered for light-dependent ($\gamma_{TLD}$) and light-independent ($\gamma_{TLI}$) emissions"

Page 4 line 26 "The light dependent factor" could be replaced by "For light-dependent emission, this activity factor..."

Page 4 line 27: same, for light-independent emissions

Page 5 line 26: replace "Sect." by "Table"

Page 5 line 32: insert a comma after "emission factors"

Page 5 line 33: "correspondence"

Page 5 same line: "classifications"

Page 6 line 9: replace "decrease" by "lies"

Page 6 line 15: delete "together"

Page 6 line 16: "correspondence"

Page 6 line 14: delete "the" after "All"

Page 6 same line: replace "south" by "southern"

Page 6 lines 20-21: suggest replacing "mainly impact" by "are the main driver for"

Page 8: the parentheses are mixed up for Misztal et al. 2015

Page 9 line 9: "Its"

Page 9 line 11: replace "emissions" by "$CO_2$ levels"

Page 10 line 8: replace "are" by "is"

Page 10 line 11 "PFT-specific emission factors calculated...": weird, please rephrase

Page 10 lines 22-23: parentheses mixed up for Guenther et al. 2012

Page 14 line 12: parentheses wrong for Zhang et al. 2014

Page 14 line 24: replace "fairly agrees" with "agrees fairly well"

Page 15 line 6: "impacts"

Page 15 line 17: replace "depending on" by "of"

Page 15 line 22: see remark above regarding interannual variability

Table 2: Please explain in the legend the meaning of the numbers shown into parentheses

[Figure]

---

## Referee Comment (RC2) · Anonymous Referee #2 · 1 Nov 2016

The presented paper describes implementation of the module for estimation of biogenic VOC emissions inside a chemistry climate model ECHAM6-HAMMOZ with links to land surface model JSBACH. The biogenic VOC module is based on the algorithm of the MEGAN model and in its reference setting the module is driven by emission factor data and PFT distributions provided by the MEGAN model. The authors went on to adapt the PFT categories from JSBACH to comply with and be usable by the biogenic VOC module. The correspondence between PFT categories from JSBACH and in the biogenic module can be a useful advantage in case of land cover experiments (past or future scenarios). The authors performed series of simulations in order to compare results of their BVOC emission module with previous studies and to address sensitivities of the emission estimates to input data (emission factors and PFT distributions, impact of soil moisture deficit and nudged meteorology). The resulting reference emissions are comparable with previously published studies and the sensitivity analysis suggests limits and uncertainties of modeled emissions. Overall, the paper presents a successful integration of an online module for estimation of biogenic VOC emissions inside the ECHAM6-HAMMOZ model system. This development allows the model to use BVOC emissions consistent with the rest of the system (e.g. using same meteorological fields, leaf area index) and provides a useful tool for simulating BVOC emissions under changing climate and/or vegetation distributions.

I find the manuscript well prepared. It is structured in a straightforward way. The methodology and results are described in very comprehensively. I therefore recommend the manuscript to be published in GMD after addressing minor comments mentioned below.

Specific comments

Section 2.3.1 Emission activity factor gamma

I believe there is a missing parenthesis at the end of Eq. (3) and the gamma_LAI factor multiplies the light independent as well as the light dependent part. The approach to calculation of the gamma factor is a combination of the algorithm used in the newest MEGANv2.1 (Guenther et al., 2012) and the simplified parameterized approach (PCEEA) described in Guenther et al. (2006) for isoprene, i.e. for light dependent species.

The authors should make clear that the Eq. (3) as it is described is actually not used by MEGANv2.1, but already includes edits after application of PCEEA. I think it would increase clarity if the authors described with a little bit more detail how they derived Eq. (3). Guenther et al. (2006) do not discuss light-dependent and light-independent parts (as the paper is focused on isoprene only) and Guenther et al. (2012) introduce

light dependent fraction factor, but do not mention the final equation for calculation of gamma_CE for both light-dependent and light-independent parts in a way as it is used in the model code (which I assume was a starting point for the presented study), therefore it is not straightforward how authors end up with Eq. (3).

I assume the construction of Eq (3) was the following and I'd suggest the authors to include its derivation (in modified way) in the manuscript.

Equation for calculation of gamma_CE in MEGANv2.1 (as written in the MEGANv2.1 code) is

Gamma_CE = (1-LDF) * gamma_TLI * gamma_LAI + LDF * Cce * LAI * gamma_TLD

Following Guenther et al. (2006) the calculation of light-dependent factor with detailed canopy environment model (i.e. Cce*LAI*gamma_TLD) was replaced by parameterized canopy environment emission activity factor (gamma_LAI*gamma_P*gamma_T)

My other comment to Eq (3) is that gamma_TLI factor (accounting for temperature dependence for light-independent species) is in MEGANv2.1 calculated for sunlit and shaded leaves at 5 canopy levels. This approach is obviously not used in the current study. I think the authors should mention the simplification they have done for calculation of the light-independent factor and eventually comment on its implications.

P7L29 : Loss of biomass (annual cycle of LAI) also contributes to seasonal variation of emissions.

P13L5: Sindelarova et al. (2014) suggested that a considerable uncertainty in applying the soil moisture activity factor lies in the wilting point value which differs among the models. Authors mention the importance of wilting point selection themselves. Could they comment on why they chose 35 % of the maximum soil water content as a wilting point value?

P14L14-15: I suppose that the reductions of 1 Tg(C)/year for isoprene and 0.08 Tg(C)/year are averages over the modeled period. It should be mentioned in the sentence that these are mean values.

P16L3-5: The authors say that "The use of emission factors derived from PFT distributions . . . results . . . to the largest changes in the spatial distribution of BVOC emissions" but it is not clear what they compare here. Largest changes compared to other simulations in the current paper?

Table 6. Please revise the unit in the Table caption

Technical comments

P3L20: please add a reference for ERA-Interim dataset

P4L28: change "classe" to "class"

P6L25: correct "nitric oxyde" to "nitric oxide"

P10L23 : The whole reference Guenther et al. (2012) should be enclosed in parenthesis

P11L9: erase "the" from "increases of all the monoterpene compounds"

P11L9-10: please rephrase "due to the presence in the major part of Australia of temperate shrub PFTs, which are strong emitters of monoterpenes" to "due to the presence of temperate shrub PFTs, strong monoterpene emitters, in major part of Australia"

P11L12: erase "the" from "to the larger spatial coverage"

P11L17 and L19: replace "oxid" with "oxide"

P11L20: erase "the" from "due to the very low"

P11L22: erase "the" from "The much higher values"

P12L3-4: please rephrase "which are replaced in the JSBACH distribution by cool/cold C3 grasses or bare soil at high latitudes of the Northern Hemisphere" with "which are replaced by cool/cold C3 grasses or bare soil at high latitudes of the Northern

Hemisphere in the JSBACH distribution"

P12L12: replace "PFT cover" by "PFT coverage"

P15L4: correct the sentence: either "decreases are" or "decrease is"

P15L6: correct: "only weakly impacts global mean"

P15L31: add "the" before "Earth"

P16L14: replace "highlight" with "highlights"

P16L15: replace "it's " by "its"

P16L17: replace "water" by "moisture"

Figure 1, 5, 8, 12: Label for the y-axis is missing

Figure 2: Labels for the x- and y-axis are missing

Figure 3, 4, 6 and 15: Numbers on colorbars are very difficult to read. They would benefit from bigger fonts.

Figure 13: Please use brackets to enclose the unit in the y-axis labels.

---

## Referee Comment (RC3) · Anonymous Referee #3 · 8 Nov 2016

General comments:

The GMDD paper by A.-J. Henrot and coworkers present a modeling work on biogenic emissions of volatile organic compounds, presenting the implementation of the MEGAN v2.1 model in the ECHAM6-HAMMOZ chemistry climate model. Several tests, changing the forcing or set up considered for the simulations, are carried out to quantify the variability and sensitivity of calculated emission estimates.

The paper is well written and the work clearly presented, sheding lights on important aspects of BVOC emission modeling. Following the other reviewers feedbacks, I have

a few additional comments and corrections, mostly minor, that I would like to be considered before publication in GMD, which I warmly support.

Specific comments:

In the abstract it is written that "Isoprene emissions show the highest sensitivity to soil moisture impact", which to me is slightly confusing as soil moisture is taken into account only for isoprene, and not for other compounds. The sentence should be modified for instance to "The highest sensitivity of isoprene emissions is calculated when considering soil moisture impact."

The leaf area index is a key driving variable in BVOC emissions, and I think a few more details should be given regarding this topic. Especially, how is the consistency between vegetation type and LAI given when switching from the 11 PFT classification to the 14 extended PFT one? Is the same LAI considered for all new categories?

Page 7, line 27: Please give more information regarding the biomass density calculation and relation with vegetation classification used, either here or preferably in the model description section.

Technical corrections:

Page 1, abstract, line 13: remove "of the" in "most of the compounds"

Page 2, line 5: remove "as well" in "BVOC emissions as well influence"

Page 2, line 22-25: the reference list is given twice, remove one

Page 6, line 6: remove "the" in "for the trees,"

Page 6, line 7: avoid "double parenthesis" for instance in "(1990-2009)" already given between parenthesis

Please check the format for references citations throughout the manuscript. For instance remove the citation year parenthesis page 6, line 6 in "based on Levis et al. GMDD
(2004)" as this sentence is already between parenthesis. Same page 10, line 17 for (... in Sinderalova et al. (2014)). Page 10, line 22-23 change "Guenther et al. (2012) to "(Guenther et al., 2012).

Page 7, line 6: change "Monoterpenes global annual emission" to "Monoterpene global annual emission"

Page 7, line 18: remove "of the" in "most of the compounds"

Page 9, line 9: change "It's impact on isoprene emission" to "Its impact on isoprene emission"

Page 9, line 14: change "It's activation increases" to "Its activation increases"

Page 12, line 16: remove "of the" in "most of the compounds"

Page 16, line 15: change "of it's impact on other compounds" to "its impact on other compounds.

GMDD

---

## Author Comment (AC1) · 30 Nov 2016

**Interactive comments on « Implementation of the biogenic emission model MEGAN(v2.1) into the ECHAM6-HAMMOZ chemistry climate model. Basic results and sensitivity tests » by Alexandra-Jane Henrot et al.**

**Alexandra-Jane Henrot et al.**

**REFEREE #1**

We would like to thank Referee #1 for useful and constructive comments which have helped us to improve the manuscript. The suggested changes will be addressed in the revised version of the manuscript.

Referee #1's comments are quoted in blue. Authors' answers are in regular font and authors' changes in the manuscript are quoted in italic.

**Major comment**

My only major objection lies is the use of a hybrid MEGAN model algorithm which mixes elements of the version 2.1 (Guenther et al., 2012) with the PCEEA developed for isoprene as part of MEGAN v2 (Guenther et al., 2006) and with the earlier algorithm of Guenther et al. (1993). The reader might want to know whether this setup provides results similar to those of the full version 2.1. Besides problems with the temperature dependence of light-independent emissions (see Specific comments, below), I am especially worried that the dependence on LAI is not correct. In Equation 3, I assume that $\gamma_{LAI}$ multiplies both ligh-dependent and light-independent terms. The expression of $\gamma_{LAI}$ is obtained from PCEEA, which is fine for light-dependent emissions, but certainly not for light-independent emissions, which are essentially proportional to LAI (cf. Equation 2 in Guenther et al., 2012). It is possible, although I'm not certain, that the authors were misled by a recently published model study (Messina et al., ACPD 15, 33967-34033, 2015). Messina et al. made the very surprising claim that, according to MEGAN, monoterpenes emissions show very little sensitivity (less than isoprene emissions) to changes in LAI. This result cannot be correct, and is contradicted by their reported results obtained with the ORCHIDEE model. Basically, the emission is proportional to the amount of leaf biomass, which is proportional to LAI. For light-dependent emissions, light attenuation dampens this relation. For light-independent emissions, this effect does not exist. Could the authors also estimate the sensitivity of their estimated isoprene and monoterpenes emissions to a given change in LAI (say, a factor of 1.5)?

According to the comments of Referees #1 and #2 we have amended the description of the calculation of the activity factor GAMMA_CE, in order to clarify the parameterizations used in the biogenic module presented in this study. As pointed

out by Referees, the calculation of the GAMMA_CE factor applied here is a combination of the parameterization used in MEGANv2.1 and of the PCEEA approach for the light-dependent compounds as described in Guenther et al. (2006). The description of GAMMA_CE calculation has been modified (see Author response to Referee #2, specific comment 1). A discussion about the use of this « hybrid » parameterization and its potential effects on the global BVOC emissions has been added in the revised manuscript in section 4.2.

*« In the biogenic module applied here the light-dependent activity factor are calculated using the Parameterized Canopy Environment Emission Activity (PCEEA) approach. This bulk canopy temperature parameterization is similar to the leaf-level temperature parameterization of the explicit canopy model but is slightly less sensitive to temperature. Guenther et al. (2006) report estimates of annual global isoprene emissions with the PCEEA approach that are within 5 % of the value estimated using the standard MEGAN canopy environment model, but differences can be up to 25 % for estimates at specific times and locations. »*

As noticed by Referee #1, the light-independent activity factor GAMMA_TLI is calculated here following the monoterpene exponential temperature response function of Guenther et al. (1993). This algorithm is similar to the algorithm used in the fortran code of MEGANv2.1 for calculating GAMMA_TLI for all compounds with a light-independent activity, and is explicitly described in Guenther et al. (2012), Section 2.2, Eq (11). The only difference introduced here is that we assume that leaf temperature is equal to ambient air temperature.  This simplification is the subject of Referee #2 second specific comment. Following his/her suggestion, we add a brief description and discussion of this simplification and implications in the revised manuscript (see Author response to Referee #2, specific comment 2).

Equation 3 has been corrected (missing parenthesis added) as follows:

GAMMA_CE = GAMMA_LAI * ((1-LDF) * GAMMA_TLI + LDF * GAMMA_TLD).

Indeed GAMMA_LAI multiplies both the light-dependent and light-independent activity factors. The first term of Eq (3) is similar to the calculation of light-independent activity factors in the fortran code of MEGANv2.1 (GAMMA_LAI*(1-LDF)*GAMMA_TLI). We agree with Referee #1 that this equation does not correspond to the development of Eq (2) in Section 2.2 in Guenther et al. (2012) for the light-independent fraction. Here, we based our model development on the MEGANv2.1 fortran code, and thus used GAMMA_LAI in the calculation of the light-independent part. We have stated explicitly in the revised manuscript that Eq (3) is derived from the basic equation used in the fortran code of MEGANv2.1 and differs from Eq (2) in Section 2.2 in Guenther et al. (2012). Messina et al. (2015) already discussed the use of GAMMA_LAI for both light-dependent and light-independent emissions and its effect on emissions.

Following the suggestion of Referee #1 we did an additional sensitivity test over the modeled period (2000-2012) to estimate the effect on isoprene and monoterpene emissions of a change in LAI (LAI scaled by a factor 1.5). The multiplication of LAI by a factor 1.5 leads to an increase of isoprene and monoterpene global emissions by 18.5 % and 16.5%, respectively, in comparison to the reference simulation. This effect is much larger than the global annual increases of isoprene and monoterpene

emissions for the same sensitivity test with MEGANv2.1 reported in Messina et al. (2015) as 6.6 % and 6 %, respectively. However, the impact of LAI change reported here is lower in comparison to the effect of changing LAI datasets in different versions of the MEGAN model, leading to about 30 % of global annual isoprene emission changes (Guenther et al., 2006, 2012). As the same parameterization for GAMMA_LAI was used in our experiment and in the simulations performed by Messina et al. (2015), the difference in the sensitivity to LAI obtained here must result from the LAI data that are used to calculate the activity factor. The LAI distributions in winter and summer we have used for the reference simulation with the biogenic emission module are shown in Figure A below. The LAI values reported here are globally lower in comparison to the LAI values used in the MEGCRULAI simulation in Messina et al. (2015), reference simulation for the 1.5*LAI sensitivity test (Figure (4), Messina et al., 2015). In Eurasia and North America, LAI reaches a maximum value of about 3 to 3.5 $m^2$ $m^{-2}$. Messina et al. (2015) report maximum values of between 4.2 and 4.9 $m^2$ $m^{-2}$ in the same regions (from Figure (4)). The lower LAI values could explain the larger effect of LAI increase obtained here. The GAMMA_LAI factor has indeed a larger increase rate for LAI lower than 5 $m^2$ $m^{-2}$ (Messina et al. 2015). Furthermore, Messina et al. (2015) reported a possible weaker impact of high LAI in MEGANv2.1 in their simulations due to the leaf self-shading effect (an increase in LAI increases the proportion of shaded and cooler leaves thus leading to lower emission rates (Sindelarova et al., 2014). This effect is not taken into account here due to the use of the PCEEA approach. We can finally remark that as both light-dependent and light-independent emissions are calculated using the GAMMA_LAI factor, isoprene and monoterpene emissions do not show significant differences in their sensitivity to LAI.

A description of this additional sensitivity test and discussion of the results obtained, have been added in the revised manuscript in Section 4.3.

[Figure]

Figure A: Leaf Area index (LAI, $m^2$ $m^{-2}$) geographical distribution for winter (DJF) and summer (JJA) in the reference simulation.

**Specific comments**

The title is long, I think that its second part (Basic results and sensitivity test) could be dropped.

We agree to shorten the title of the paper and to remove the second part of the title. The revised manuscript will be entitled *"Implementation of the biogenic emission model MEGAN(v2.1) into the ECHAM6-HAMMOZ chemistry climate model."*

Page 3 line 30: please provide some more details on those global potential land cover maps. In what sense are those for potential land cover? Do they include realistic representation of human influence (e.g. crops, managed forests, etc.)?

The background of the land cover map used here is a map of potential vegetation derived from the reconstruction of Ramankutty and Foley (1999). Potential vegetation means here the vegetation that would exist in the climax state under today's conditions and in the absence of human activities. The potential vegetation map is then combined with land use maps (agricultural types considered are croplands, C3, and C4 pastures) of Ramankutty and Foley (1999) and Foley et al. (2003). The land cover map used here thus takes into account to some extent the human influence on the global vegetation distribution. We refer to Pongratz et al. (2008) for the details of the land cover map construction.

Page 4 line 25: the common MEGAN assumption that LDF is a species-specific constant is very unrealistic for monoterpenes (e.g. Rinne, Atmos. Chem. Phys. Discuss., 15, C11977–C11979, 2016). This should be (at least) mentioned somewhere in the manuscript.

As suggested by Referee #1, we have mentioned in the revised manuscript (in Section 4.2) the uncertainties linked to the use of species-specific light-dependent factors in the MEGAN model. The text has been amended as follows:

*"The introduction of light-dependent factors for other compounds than isoprene in MEGANv2.1 has notably a significant effect on global emissions (Messina et al., 2015). Moreover, a large range of variation of light-dependent emissions, especially for monoterpenes, is observed across plant species (Rinne, 2016). Thus, the use of a single LDF value per compound in MEGANv2.1 can introduce further uncertainties in the model emission estimates and discrepancies between model versions using different values of LDF."*

Page 4 line 26: the temperature activity factor for light-dependent emissions is obtained from G06 (Guenther et al., 2006) but I find several differences between G06 and the expressions given in the supplement, e.g. the factor $C_{eo}$ is equal to 1.75 for isoprene in G06. Please explain.

The temperature activity factor for light dependent emission GAMMA_T is calculated as described in Guenther et al. (2006). The factor Ceo has been updated to the value used in MEGANv2.1 in order to be consistent with the values of Ceo used for the other compound and taken from MEGANv2.1.

Page 4 line 28: this temperature dependence (Guenther et al. 1993) is considerably simpler than the temperature activity factor of Guenther et al. 2012. Please discuss the possible implications of this simplification.

See response to major comment

Page 7 line 14: the globemission website does not provide the box locations. The latitudes/longitudes of the regions should be specified in the figure legend.

As suggested by Referee #1, the latitudes/longitudes of the regions selected will be specified in the legend of Figure (1). This information is indeed not directly available from the website of the GlobEmission project.

Figures 3, 4, 6, 15: please enlarge the fonts, or enhance resolution for better readability

Figures have been enlarged to allow a better readability of the legends.

Page 8, lines 30-32: Messina et al. fail to provide good reasons for this supposed lower sensitivity of BVOC emissions to LAI in MEGANv2.1. The ORCHIDEE sensitivity to LAI makes much more sense.

This comment has been addressed in the response to the major comment.

Page 13 lines 2-3: here the soil water activity factor is said to depend on relative soil water amount. But the supplement reports a dependence on volumetric water content, as in G06. Are those two quantities the same thing?

The soil moisture content can be expressed as the amount of water (in m of water depth) present in the soil (depth of the soil water reservoir) or also in percent of volume (volume of water in volume of soil water reservoir). Both quantities are relatives and represent the same values if the surface of the soil water reservoir considered (here the surface of the grid-cell) is the same. We have modified Eqs (S10) to (S13) in the supplement to mention the relative water amount instead of the volumetric water amount in the calculation of GAMMA_SM.

Page 14 lines 21-24: Interannual variability is not well quantified by the ratio of maximum and minimum values. Since the periods covered by the different studies are all different, I'm afraid that the comparison amounts to a comparison of apples with oranges. It would make more sense to compare the standard deviation of annual totals in the different datasets.

Following the suggestion of Referee #1, we have based our comparison of interannual variability on the standard deviations of annual total isoprene emissions obtained here and calculated from the available information given in Sindelarova et al. (2014), Müller et al. (2008) and Lathière et al. (2006). The text of the revised manuscript in Section 4.3.4 has been amended as follows:

*"The standard deviation of total annual isoprene emissions obtained here (+/- 9.1 Tg C/yr) is lower than the standard deviations of total annual isoprene emissions of +/- 30 Tg C/yr and +/- 20.2 Tg C/yr reported by respectively Müller et al. (2008) (1995-2006 MEGANv2 simulation forced with ECMWF reanalysis) and Sindelarova et al. (2014) (1980-2010 MEGANv2.1 simulation forced with MERRA reanalysis). However, the standard deviation of the reference simulation is closer to the +/- 10.8 Tg C/yr standard deviation obtained by Lathière et al. (2006) for a 1983-1995 simulation using the MEGANv2 model forced with satellite based climate archive."*

Page 15 lines 8-9: I suppose that the process working on here is the influence of precipitation on soil moisture and hence on $\gamma_{SM}$. If so, this should be stated explicitly.

Indeed, the process described here is the decrease of isoprene emission in response to a reduction of soil moisture due to the decrease of precipitation obtained in the TEST_NUDG+SM simulation. The corresponding sentence has been amended as suggested to mention explicitly this process.

**Technical comments**

We have taken into account all technical corrections suggested by Referee #1.

**References**

Foley, J. A., Delire, C., Ramankutty, N., and Snyder, P : Green Surprise? How terrestrial ecosystems could affect earth's climate, Front. Ecol. Environ., 1:38-44, 2003.

Guenther, A. B., Zimmerman, P. R., Harley, P. C., Monson, R. K., and Fall, R.: Isoprene and monoterpene emission rate variability: Model evaluations and sensitivity analyses, J. Geophys. Res., 98, 12 609-12 617, 1993.

Guenther, A. B., Karl, T., Harley, P., Wiedinmyer, C., Palmer, P. I., and Geron, C.: Estimates of global terrestrial isoprene emissions using MEGAN (Model of Emissions of Gases and Aerosols from Nature), Atmos. Chem. Phys., 6, 3181-3210, 2006.

Guenther, A. B., Jiang, X., Heald, C. L., Sakulyanontvittaya, T., Duhl, T., Emmons, L. K., and Wang, X.: The Model of Emissions of Gases and Aerosols from Nature version 2.1 (MEGAN2.1): an extended and updated framework for modeling biogenic emissions, Geosci. Model Dev., 5, 1471-1492, 2012.

Lathière, J., Hauglustaine, D. A., Friend, A. D., Noblet-Ducoudré, N. D., Viovy, N., and Folberth, G. A.: Impact of climate variability and land use changes on global biogenic volatile organic compound emissions, Atmos. Chem. Phys., 6, 2129-2146, 2006.

Messina, P., Lathière, J., Sindelarova, K., Vuichard, N., Granier, C., Ghattas, J., Cozic, A., and Hauglustaine, D. A.: Global biogenic volatile organic compound emissions in the ORCHIDEE and MEGAN models and sensitivity to key parameters, Atmos. Chem. Phys. Discuss., 15, 33 967-34 033, 2015.

Müller, J.-F., Stavrakou, T., Wallens, S., Smedt, I. D., Roozendael, M. V., Potosnak, M. J., Rinne, J., Munger, B., Goldstein, A., and Guenther, A. B.: Global isoprene emissions estimated using MEGAN, ECMWF analyses and a detailed canopy environment model, Atmos. Chem. Phys., 8, 1329-1341, 2008.

Pongratz, J., Reick, C., Raddatz, T., and Claussen, M.: A Global Land Cover Reconstruction AD 800 to 1992 - Technical Description, Tech. Rep. 51, Max-Planck-Institut für Meteorologie, Hamburg, 2008.

Ramankutty, N. and Foley, J. A.: Estimating historical changes in global land cover: croplands from 1700 to 1992, Global Biogeochem. Cycles, 13, 997-1027, 1999.

Rinne, J. : Interactive comment on "Global biogenic volatile organic compound emissions in the ORCHIDEE and MEGAN models and sensitivity to key parameters" by P. Messina et al., Atmos. Chem. Phys. Discuss., 15, C11977-C11979, 2016

Sindelarova, K., Granier, C., Bouarar, I., Guenther, A., Tilmes, S., Stavrakou, T., Müller, J.-F., Kuhn, U., Stefani, P., and Knorr, W.: Global data set of biogenic VOC emissions calculated by the MEGAN model over the last 30 years, Atmos. Chem. Phys., 14, 9317-9341, 2014.

---

## Author Comment (AC2) · 30 Nov 2016

Interactive comments on « Implementation of the biogenic emission model MEGAN(v2.1) into the ECHAM6-HAMMOZ chemistry climate model. Basic results and sensitivity tests » by Alexandra-Jane Henrot et al.

Alexandra-Jane Henrot et al.

**REFEREE #2**

We would like to thank Referee #2 for positive and constructive comments on our paper, and especially for the suggestions made for clarifying the calculation of the activity factors. The suggested changes will be addressed in the revised version of the manuscript.

Referee #2's comments are quoted in blue. Authors' answers are in regular font and authors' changes in the manuscript are quoted in italic.

**Specific comments**

**Section 2.3.1 Emission activity factor gamma**

I believe there is a missing parenthesis at the end of Eq. (3) and the gamma\_LAI factor multiplies the light independent as well as the light dependent part. The approach to calculation of the gamma factor is a combination of the algorithm used in the newest MEGANv2.1 (Guenther et al., 2012) and the simplified parameterized approach (PCEEA) described in Guenther et al. (2006) for isoprene, i.e. for light dependent species.

The authors should make clear that the Eq. (3) as it is described is actually not used by MEGANv2.1, but already includes edits after application of PCEEA. I think it would increase clarity if the authors described with a little bit more detail how they derived Eq. (3). Guenther et al. (2006) do not discuss light-dependent and light-independent parts (as the paper is focused on isoprene only) and Guenther et al. (2012) introduce light dependent fraction factor, but do not mention the final equation for calculation of gamma\_CE for both light-dependent and light-independent parts in a way as it is used in the model code (which I assume was a starting point for the presented study), therefore it is not straightforward how authors end up with Eq. (3).

I assume the construction of Eq (3) was the following and I'd suggest the authors to include its derivation (in modified way) in the manuscript.

Equation for calculation of gamma\_CE in MEGANv2.1 (as written in the MEGANv2.1 code) is

Gamma\_CE = (1-LDF) \* gamma\_TLI \* gamma\_LAI + LDF \* Cce \* LAI \* gamma\_TLD

Following Guenther et al. (2006) the calculation of light-dependent factor with

detailed canopy environment model (i.e. Cce\*LAI\*gamma\_TLD) was replaced by parameter- ized canopy environment emission activity factor (gamma\_LAI\*gamma\_P\*gamma\_T)

Equation 3 has been corrected (missing parenthesis added) as follows:

GAMMA\_CE = GAMMA\_LAI \* ((1-LDF) \* GAMMA\_TLI + LDF \* GAMMA\_TLD).

As pointed out by Referee #2, the calculation of GAMMA\_CE used in the biogenic model described here differs from the basic calculation applied in MEGANv2.1. The light-dependent activity factor GAMMA\_TLD is indeed calculated using the PCEEA approach (Guenther et al., 2006), which is not the case in the equation used in MEGANv2.1. Accordingly to the suggestion of Referee #2, we have added in the revised manuscript in Section 2.3.1 a detailed description of the construction of Eq (3).

« GAMMA\_CE accounts for variations associated with Leaf Area Index (LAI)  $(m^2m^{-2})$ , Photosynthetic Photon Flux Density (PPFD) (µmol of photons in 400-700 nm range  $m^{-2} s^{-1}$ ) and temperature (K). The basic equation used in the fortran code of MEGANv2.1 to calculate GAMMA\_CE is:

GAMMA\_CE = GAMMA\_LAI\*(1-LDF)\*GAMMA\_TLI + Cce\*LAI\*LDF\*GAMMA\_TLD

where Cce is the canopy environment coefficient (assigned a value that results in GAMMA = 1 for the standard conditions), and GAMMA\_LAI, GAMMA\_TLI and GAMMA TLD are the activity factors for LAI, light and temperature. Different expressions for the activity factor for temperature are considered for light dependent (GAMMA\_TLD) and light independent (GAMMA\_TLI) emissions using the light dependence fraction LDF specific for each compound (Guenther et al., 2012). Light dependent emissions are calculated following the isoprene-response to temperature described in Guenther et al. (2006). Light independent emissions follow the monoterpene exponential temperature response described in Guenther et al. (1993). In order to avoid the use of a detailed canopy environment model calculating light and temperature at each canopy depth, we applied the Parameterised Canopy Environment Emission Activity (PCEEA) approach (Guenther et al., 2006). The calculation of the light-dependent activity factor with a detailed canopy environment model (i. e. Cce\*LAI\*GAMMA TLD) is replaced by a parameterized canopy environment activity factor (i. e. GAMMA LAI\*GAMMA P\*GAMMA T) as described in Guenther et al. (2006). We refer the reader to the description of Guenther et al. (2006, 2012) for the details of the LAI and light-dependent activity factor computations. Detailed formula and parameters per compound class are given in Supplementary Material (Sect. S1 and S2). The equation for GAMMA\_CE applied here is thus:

GAMMA\_CE = GAMMA\_LAI\*((1-LDF)\*GAMMA\_TLI + LDF\*GAMMA\_P\*GAMMA\_T).

My other comment to Eq (3) is that gamma\_TLI factor (accounting for temperature dependence for light-independent species) is in MEGANv2.1 calculated for sunlit and shaded leaves at 5 canopy levels. This approach is obviously not used in the current study. I think the authors should mention the simplification they have done for calculation of the light-independent factor and eventually comment on its implications.

In agreement with the second comment of Referee #2 we have mentioned in the revised manuscript in Section 2.3.1 the simplification applied for the calculation of light-independent activity factor GAMMA\_TLI and briefly discussed its implication. In the biogenic module we use the ambient air temperature instead of the leaf temperature to calculate GAMMA\_TLI. This simplification was also applied in Guenther et al. (1995). We thus do not take into account in BVOC emission estimates the effect of the difference of temperature between sunlit and shaded leaves. Leaves in direct sunlight often experience temperatures that are a degree or more higher than ambient air while shaded leaves are often cooler than ambient air temperature (Guenther et al., 2012). This simplification leads to a small underestimation (<5%) of light-independent BVOC emissions accordingly to Guenther et al. (2012). The text has been completed as follows:

« The light-independent activity factor GAMMA\_TLI is calculated here assuming that leaf temperature is equal to ambient air temperature. In the absence of a detailed canopy model, we do not distinguish between sunlit and shaded leaves that can show significant temperature differences. Leaves in direct sunlight often experience temperatures that are a degree or more higher than ambient air while shaded leaves are often cooler than ambient air temperature (Guenther et al., 2012). This simplification can lead to a small underestimation (< 5 %) of light-independent BVOC emissions as reported in Guenther et al. (2012). »

P7L29 : Loss of biomass (annual cycle of LAI) also contributes to seasonal variation of emissions.

The loss of biomass has been mentioned in the corrected sentence.

P13L5: Sindelarova et al. (2014) suggested that a considerable uncertainty in applying the soil moisture activity factor lies in the wilting point value which differs among the models. Authors mention the importance of wilting point selection themselves. Could they comment on why they chose 35 % of the maximum soil water content as a wilting point value?

The soil water content and wilting point used in the biogenic module are calculated in the soil water model included in ECHAM6 (Hagemann and Stacke, 2002; Hagemann and Stacke, 2015). In the soil water model, the permanent wilting point is set to 35% of the maximum soil water amount and we stick to this value in order to be consistent with the soil water model of ECHAM6.

P14L14-15: I suppose that the reductions of 1 Tg(C)/year for isoprene and 0.08 Tg(C)/year are averages over the modeled period. It should be mentioned in the sentence that these are mean values.

As suggested, we have modified the sentence to mention that the values reported are means over the modeled period.

P16L3-5: The authors say that "The use of emission factors derived from PFT distributions . . . results . . . to the largest changes in the spatial distribution of BVOC emissions" but it is not clear what they compare here. Largest changes compared to other simulations in the current paper?

The effect of the emission factors is indeed compared here to the sensitivity simulation results of the current study. The sentence has been amended to include this precision.

**Table 6. Please revise the unit in the Table caption**

The units have been corrected to  $10^{12}$  m2 in the table caption.

**Technical comments**

We have taken into account all technical corrections suggested by Referee #2.

**References**

Guenther, A. B., Zimmerman, P. R., Harley, P. C., Monson, R. K., and Fall, R.: Isoprene and monoterpene emission rate variability: Model evaluations and sensitivity analyses, J. Geophys. Res., 98, 12 609-12 617, 1993.

Guenther, A. B., Hewitt, C. N., Erickson, D., Fall, R., Geron, C., Graedel, T., Harley, P., Klinger, L., Lerdau, M., Mckay, W. A., Pierce, T., Scholes, B., Steinbrecher, R., Tallamraju, R., Taylor, J., and Zimmerman, P.: A global model of natural volatile organic compound emissions, J. Geophys. Res., 100, 8873-8892, 1995.

Guenther, A. B., Karl, T., Harley, P., Wiedinmyer, C., Palmer, P. I., and Geron, C.: Estimates of global terrestrial isoprene emissions using MEGAN (Model of Emissions of Gases and Aerosols from Nature), Atmos. Chem. Phys., 6, 3181-3210, 2006.

Guenther, A. B., Jiang, X., Heald, C. L., Sakulyanontvittaya, T., Duhl, T., Emmons, L. K., and Wang, X.: The Model of Emissions of Gases and Aerosols from Nature version 2.1 (MEGAN2.1): an extended and updated framework for modeling biogenic emissions, Geosci. Model Dev., 5, 1471-1492, 2012.

Hagemann, S. : An improved land surface parameter dataset for global and regional climate models, MPI Report 336, Max-Planck-Institut für Meteorologie, Hamburg, 2002.

Hagemann, S. and Stacke, T.: Impact of the soil hydrology scheme on simulated soil moisture memory, Clim. Dyn., 44, 1731, 2015.

---

## Author Comment (AC3) · 30 Nov 2016

**Interactive comments on « Implementation of the biogenic emission model MEGAN(v2.1) into the ECHAM6-HAMMOZ chemistry climate model. Basic results and sensitivity tests » by Alexandra-Jane Henrot et al.**

**Alexandra-Jane Henrot et al.**

**REFEREE #3**

We thank Referee #3 for insightful and constructive comments on our paper, which have helped us to improve the manuscript. The suggested changes will be addressed in the revised version of the manuscript.

Referee #3's comments are quoted in blue. Authors' answers are in regular font and authors' changes in the manuscript are quoted in italic.

**Specific comments**

In the abstract it is written that "Isoprene emissions show the highest sensitivity to soil moisture impact", which to me is slightly confusing as soil moisture is taken into account only for isoprene, and not for other compounds. The sentence should be modified for instance to "The highest sensitivity of isoprene emissions is calculated when considering soil moisture impact."

We agree with Referee #3 that this sentence can be confusing. We have amended the sentence as suggested.

The leaf area index is a key driving variable in BVOC emissions, and I think a few more details should be given regarding this topic. Especially, how is the consistency between vegetation type and LAI given when switching from the 11 PFT classification to the 14 extended PFT one? Is the same LAI considered for all new categories?

According to the comments of Referees #1 and #2, we have added in the revised manuscript a detailed description of the calculation of the activity factor depending on LAI and its impact on emission estimates, as well as a discussion of the effect of changing LAI in an additional sensitivity test.

The LAI used in the biogenic module is the LAI of the grid-cell that is directly derived from the JSBACH model (Section 2.3.1, page 5, lines 1-3). The LAI is calculated in function of climatic conditions (temperature, soil moisture) and Net Primary Productivity for several phenology types (summergreen, evergreen, raingreen, grasses and crops). It is constrained by a maximum LAI value and a specific leaf area (leaf area per gram of leaf carbon) that are PFT-specific. The PFTs are thus merged into these broad phenology types for the LAI calculation.

When switching to the 14 extended PFTs, we kept the original 11 PFT-specific parameters for the LAI calculation, in order to not modify the standard PFT classification and setup used in JSBACH. Thus, switching to the 14 extended PFT classification only allows here a better representation of PFT-specific emission factors but does not affect the LAI nor the original PFT fractions calculated in JSBACH. This clarification has been added in the revised manuscript, in Section 2.3.2.

We agree with Referee #3 that to be fully consistent in terms of vegetation dependent parameters it would be better to use the refined PFT classification directly in the JSBACH model in order to calculate the PFT fractions and LAI. But this would require to modify the basic setup of the JSBACH model (providing the full list of specific parameters for each PFT of the new classification) and to re-calibrate the vegetation model in its full-mode (this needs very long simulations to allow the carbon stocks to reach equilibrium). This point could be addressed in a future study about the impact of land cover and land-use change on the atmospheric chemistry within the ECHAM6-HAMMOZ model.

Page 7, line 27: Please give more information regarding the biomass density calculation and relation with vegetation classification used, either here or preferably in the model description section.

The sentence pointed out here is a general sentence about the potential of emission of tropical regions. In the present model, the variation of the biomass is calculated by the vegetation model JSBACH and taken into account for the biogenic emissions via the corresponding changes in LAI. This precision has been added in the revised manuscript in Section 2.3.1.

**Technical comments**

We have taken into account all the technical corrections suggested by Referee #3.

---

## Author Response (AR2)

**Author comment replying to the second report of Referee #1.**

**Alexandra-Jane Henrot et al.**

We would like to thank Referee #1 for the second report commenting the revised manuscript.

Referee #1's comment is quoted in blue. Authors' answer is in regular font and authors' changes in the manuscript are quoted in italic.

I thank the authors for the clarifications and additional sensitivity calculations in response to my comments.
Still, regarding the algorithm description (Eq. (3)), the authors response says that the manuscript now states explicitly that Eq (3) differs from Eq (2) in Guenther et al. (2012). I don't see this information anywhere in the manuscript. I think this is important. The LAI activity factor from PCEEA is not appropriate for light-indenpendent emissions. This should be made clear somewhere in the paper.

Following the suggestion of Referee #1, we have amended the algorithm description in Section 2.3.1 in order to state explicitly that Eq. (3) is different from the general equation for activity factors (Eq. (2)) given in Guenther et al. (2012). A clarification has been added as follows:

[revised manuscript text omitted]